# AN AUDITING TEST TO DETECT BEHAVIORAL SHIFT IN LANGUAGE MODELS

**Leo Richter**[1]     **Xuanli He**[1]     **Pasquale Minervini**[2,3]     **Matt J. Kusner**[4,5]

[1]UCL Centre for Artificial Intelligence, University College London, United Kingdom
[2]School of Informatics, University of Edinburgh, United Kingdom
[3]Miniml.AI, United Kingdom
[4]Polytechnique Montréal, Canada
[5]Mila – Quebec AI Institute, Canada
`ucablri@ucl.ac.uk, matt.kusner@mila.quebec`

## ABSTRACT

As language models (LMs) approach human-level performance, a comprehensive understanding of their behavior becomes crucial to avoid potential harms. While extensive initial evaluations, including red teaming and diverse benchmarking, can establish a behavioral profile, subsequent fine-tuning or deployment modifications may alter these model behaviors in unintended ways. We study the *behavioral shift auditing* problem, where the goal is to detect unintended changes in model behavior. We formalize this problem as a sequential hypothesis test. We apply and extend a recent testing method to include a configurable tolerance parameter that adjusts sensitivity to behavioral changes for different use cases. The test is guaranteed to be consistent and has tight control over the Type I error rate. We evaluate our approach using two case studies: monitoring model changes in (a) toxicity and (b) translation performance. We find that the test is able to detect distribution changes in model behavior using hundreds of prompts.

## 1 INTRODUCTION

Language models (LMs) can now achieve human-level performance in a wide range of tasks, including text summarization, machine translation, coding, and even acting as AI scientists: generating hypotheses and designing experiments (Achiam et al., 2023; Katz et al., 2024; Lu et al., 2024; Zhang et al., 2024). As capabilities continue to scale, evaluating LM behaviors becomes increasingly important and increasingly difficult (Hendrycks et al., 2021; Ngo et al., 2022; Wolf et al., 2023). Large-scale evaluations—such as comprehensive behavior and capability assessments (Wang et al., 2023a) and red-teaming exercises (Perez et al., 2022a)—are widely used to verify that language models (LMs) behave safely and as expected. However, these evaluations tend to be expensive and are not well-suited for continuous monitoring, especially when models are updated or fine-tuned with new data. This is problematic because even seemingly benign or narrow modifications can inadvertently lead to undesirable changes in model behavior (Qi et al., 2023; Betley et al., 2025). This raises the question: How can we quickly and cheaply detect unwanted changes in LM behavior?

Consider two hypothetical settings where this question might be asked: (1) *Internal Audit*: A company develops a language model that has passed rigorous safety and performance evaluations. After deploying the model, they continue to fine-tune it to improve its performance on certain tasks. The development team wants to stay informed about any *drastic* changes this might induce in the model's behavior—particularly shifts in areas unrelated to the intended updates. How can the team rapidly detect meaningful changes in model behavior throughout the development cycle? (2) *External Audit*: A regulatory body certifies a language model for public deployment after extensive safety evaluation. However, they are concerned that the deployed model's behavior may change over time due to updates or intentional modifications. Since they only have access to the model through an API and cannot inspect its internal parameters, they require a mechanism to regularly check that the model's behavior remains consistent with the certified version. How can the regulator regularly check

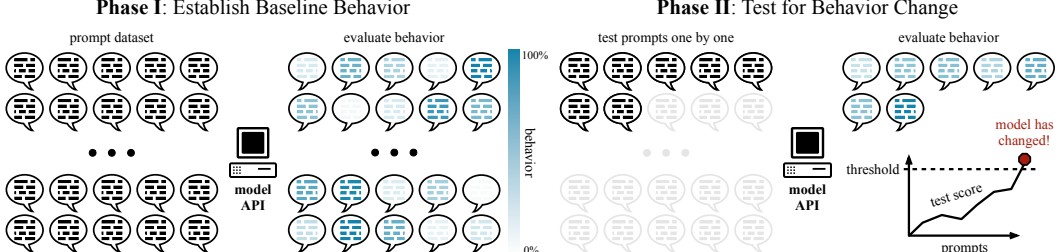

Figure 1: Overview of behavior shift auditing framework.

the deployed model's behavior is the same as the previously certified one? We call the problem of detecting changes in LM behavior distributions over time *behavioral shift auditing* problems.

In this paper, we formalize the problem of behavioral shift auditing in language models and propose a statistical test that monitors changes in model behavior using only black-box access (e.g., via API calls). Our goal is to develop a sample-efficient method that guarantees detection of behavioral shifts while tightly controlling the rate of false positives. Further, it should provide the user with a tolerance parameter that allows a behavior distribution to change by some amount $\epsilon$ without triggering a detection. This parameter controls the strictness of the auditing test - in some settings (e.g., example (1)), a more liberal $\epsilon$ might be appropriate, while in other cases (e.g., example (2)) one might require a more conservative $\epsilon$ or even want to disallow any change at all. The key insight behind our approach is to frame behavior shift auditing as a hypothesis testing problem over the model's behavior distribution. This framing makes our method applicable to a wide range of measurable behaviors—such as dangerous capabilities (Phuong et al., 2024), mathematical reasoning (Mishra et al., 2022a), and biases (Wang et al., 2023a; Kotek et al., 2023). To this end, we leverage and extend recent advances in testing by betting (Pandeva et al., 2024). Under mild assumptions, our sequential test provably detects any change given enough samples, while ensuring non-asymptotic control over false positives. We demonstrate our test on detecting shifts in toxicity and translation performance. We find that we can detect changed LM behaviors using hundreds of prompts. We release our code here: `https://github.com/richterleo/lm-auditing-test`.

## 2 RELATED WORK

**LM behavior functions.**    Early evaluations of NLP models relied on curated datasets for detecting biases or toxicity (Bolukbasi et al., 2016); larger collections of data were constructed e.g. through web scraping (Zhao et al., 2018; Zampieri et al., 2019; Nangia et al., 2020; Rosenthal et al., 2021) and, more recently, by leveraging LLMs themselves to generate data (Zhang et al., 2022; Perez et al., 2023). Meanwhile, early work on behavior functions focused on measuring bias, toxicity, and hallucinations (Vidgen et al., 2020; Achiam et al., 2023; Anil et al., 2023; Chern et al., 2023; Varshney et al., 2023; Llama-team, 2024). Since the rise of LMs with human-level performance, the set of behavior functions has exploded (Zou et al., 2023b). It has become more nuanced, including complex characteristics such as power-seeking behavior (Park et al., 2023; Sharma et al., 2023), situational awareness (Zou et al., 2023a), and deception (Hagendorff, 2024). However, even with access to massive datasets and carefully constructed behavior functions it can be difficult to discover these behaviors from static inputs (Kalin et al., 2020). To address this, Perez et al. (2022a) introduced the notion of *red-teaming* for LM alignment. This allows prompts to be adversarially-constructed to expose failure cases, which arise in many state-of-the-art models (Chao et al., 2023).

**Model change identification.**    For the case where one wishes to identify *any change* in model behavior (i.e., $\epsilon = 0$) there are multiple other techniques that can be used. The first set uses ideas from *formal verification* to ensure that the predictions from a model are guaranteed to come from a specific model (Ghodsi et al., 2017; Dong et al., 2021; Fan et al., 2023; Weng et al., 2023). In general, however, these methods are computationally intensive and do not scale to state-of-the-art LMs. A second, more efficient idea is to *watermark* the model (Zhu et al., 2018; Amrit & Singh, 2022; He et al., 2022a;b; Kirchenbauer et al., 2023; Kuditipudi et al., 2023; Yoo et al., 2023). The idea is to embed signals into model generations that can be detected algorithmically. However, watermarks

are often inserted by the model owner (Kirchenbauer et al., 2023; Kuditipudi et al., 2023), allowing them (or an actor that has compromised the model) to insert it into any model that is being audited. This precludes its use for many external auditing settings. For internal auditing, a watermark may break under a small model change that is acceptable. Our work is also related to work on concept drift (Bayram et al., 2022) and prompt stability (Li et al., 2024). In principle our test can be used to detect concept and generation changes, however the focus of these works is on model performance and generation similarity, as opposed to behavior change.

**Sequential hypothesis testing.** Sequential hypothesis testing allows one to analyze data without fixing the sample size in advance (Wald, 1945), offering the potential for greater sample efficiency when significant effects exist (Arrow et al., 1949). However, naive repeated testing can increase the Type I error rate (i.e., false positives) as the number of tests grows (Jennison & Turnbull, 1999). To prevent this inflation of false positives, various methods have been developed, including the recent *testing by betting* framework (Robbins, 1970; Ramdas et al., 2023), which preserves statistical efficiency while tightly controlling the Type I error rate. Within this framework, a method called Deep Anytime-Valid Testing (DAVT) (Pandeva et al., 2024) designs powerful sequential non-parametric tests by integrating deep learning models into the testing by betting framework. They demonstrate, on a variety of tasks, including two-sample testing, competitive performance compared to other state-of-the-art non-parametric sequential tests, such as the E-C2ST (Lhéritier & Cazals, 2018) and Seq-IT (Podkopaev & Ramdas, 2024). DAVT uses a model, trained on past observations, to produce an optimized betting score on new data. In this work, we will extend this test to include a tunable tolerance parameter $\epsilon$.

## 3 PRELIMINARIES

**Testing by betting.** The testing by betting framework represents evidence against the null hypothesis as the gain in wealth $W$ of a bettor wagering on observed samples (Shafer, 2021). Before observing new samples, the bettor "buys" a test statistic at the "price" of its expected value under $\mathbf{H_0}$. After new samples are obtained, the bettor's wealth $W$ is multiplied by the ratio between the actual observed test statistic and its expectation. This ratio is referred to as the *betting score* $S_t$. The bettor reinvests in subsequent "rounds" (i.e., as new data is observed), and the observed betting scores are repeatedly multiplied, leading to a cumulative wealth process. Under $\mathbf{H_0}$, no betting strategy can consistently increase the bettor's wealth, ensuring control over the Type I error rate (Ramdas et al., 2023).

Let the bettor's (non-negative) wealth after $t$ (batches of) observations be $W_t$. In order to design a test from this wealth process we require that $W_t$ satisfies the following

$$\sup_{P \in \mathbf{H_0}} \mathbb{E}_P[W_t] \leq 1 \quad \text{for every } t \geq 0. \tag{1}$$

All non-negative stochastic processes $W_t$ that satisfy the above condition are called an **e-process** for $\mathbf{H_0}$ (Howard et al., 2021). This states that the maximum wealth across all bets cannot exceed 1 if the null hypothesis $\mathbf{H_0}$ is true.[1] Given an e-process, the test is constructed as follows: reject the null $\mathbf{H_0}$ at some time $\tau$ if $W_\tau \geq \gamma$, where $\gamma = \alpha^{-1}$ is a threshold defined by a desired significance level $\alpha \in (0, 1)$. Under $\mathbf{H_0}$, the e-process $W_t$ controls the Type I error rate. By Ville's inequality (Ville, 1939), we have:

$$\mathbb{P}_{\mathbf{H_0}} \left( \sup_{t \geq 0} W_t \geq \gamma \right) \leq \frac{1}{\gamma} = \alpha. \tag{2}$$

This ensures that the probability of incorrectly rejecting $\mathbf{H_0}$ is at most $\alpha$ at any time step. Thus, the sequential test is *anytime-valid*, maintaining error control at any stopping point.

## 4 DETECTING BEHAVIOR CHANGES

We propose an anytime-valid test for behavior shift auditing that has guarantees on its false positive rate and is consistent under certain weak assumptions. Building upon the two-sample variant of DAVT (Pandeva et al., 2024), our test introduces a customizable tolerance parameter $\epsilon$ that allows users

---

[1]It can be shown that the wealth process $W_t$ defined this way is equivalent to the minimum wealth a bettor can obtain across all $P \in \mathbf{H_0}$ (Ramdas et al., 2023).

to specify what constitutes a practically significant difference between distributions, accommodating small, insignificant variations. While we concentrate here on its application for behavior shift auditing, it may be of independent interest to the sequential hypothesis testing community. We describe the test in full generality in Appendix B.

### 4.1 AUDITING TEST

Let $X$ be a random variable representing a prompt, $\mathcal{X}$ the set of possible prompts, and $\mathbf{x} \in \mathcal{X}$ a realization of $X$. A language model is a stochastic operator $M$ that maps prompts $\mathbf{x}$ to generations $\mathbf{y}$. A behavior scoring function $B$ is a stochastic operator that takes a prompt and generation as input[2] and produces a score $B(\mathbf{x}, \mathbf{y}) \in [0, 1]$ (Perez et al., 2023; Wolf et al., 2024). The behavior function, prompts, and language model induce a *behavior distribution* $P_B^M$ over behavior scores $B(X, M(X))$. We can now frame the question of whether the behavior of a model $M'$ has changed (substantially) relative to a baseline model $M$ as a testing problem:

$$\mathbf{H_0}: \mathcal{D}\big(P_B^M, P_B^{M'}\big) \le \epsilon \quad \text{vs.} \quad \mathbf{H_1}: \mathcal{D}\big(P_B^M, P_B^{M'}\big) > \epsilon, \tag{3}$$

where $\epsilon \ge 0$ is a tolerance parameter, and $\mathcal{D}$ is a distance measure between probability distributions. Note that equality in the null hypothesis in eq. (3) corresponds to DAVT (Pandeva et al., 2024). To extend this to the composite case, our goal is to *construct an appropriate wealth process $W_t$*. This will allow us to establish error rate and consistency guarantees. To do so, we will define a betting score $S_t$ such that it produces a wealth process $W_t$ that is an e-process i.e., it satisfies eq. (1). This, in turn, will depend on the distance measure $\mathcal{D}$ that we choose.

Given a batch of prompts $x_1, \ldots, x_b$ and the distance threshold $\epsilon$ from Equation (3), we propose the *betting score*

$$S_t = \prod_{i=1}^{b} \left( \frac{1 + \phi_{t-1}\big(B(x_i, M(x_i))\big) - \phi_{t-1}\big(B(x_i, M'(x_i))\big)}{\exp(\epsilon)} \right). \tag{4}$$

where $\phi_{t-1}$ is a neural network trained on all $(t-1)$ previous batches to optimize the objective

$$\max_{\phi} \mathbb{E}[\log\left(1 + \phi(B(X, M(X))) - \phi(B(X, M'(X)))\right)].$$

Given the betting score $S_t$, we define the *wealth process* $\{W_t\}_{t \ge 1}$ of a bettor by initializing their wealth as $W_0 = 1$ and updating

$$W_t = W_{t-1} \times S_t. \tag{5}$$

If the betting score $S_t$ is an e-variable, meaning that $\mathbb{E}_{\mathbf{H_0}}[S_t] \le 1$, then the wealth process $\{W_t\}_{t \ge 0}$ is an e-process, which we can prove by induction. Under $\mathbf{H_0}$, and for any fixed $P_B^M, P_B^{M'}$ satisfying $\mathcal{D}_\Phi(P_B^M, P_B^{M'}) \le \epsilon$, $W_{t-1}$ and $S_t$ are independent. Therefore,

$$\begin{aligned} \mathbb{E}_{\mathbf{H_0}}[W_t] &= \mathbb{E}_{\mathbf{H_0}}[W_{t-1} \times S_t] \\ &= \mathbb{E}_{\mathbf{H_0}}[W_{t-1}] \times \mathbb{E}_{\mathbf{H_0}}[S_t] \le \mathbb{E}_{\mathbf{H_0}}[W_{t-1}], \end{aligned}$$

By induction, $\mathbb{E}_{\mathbf{H_0}}[W_t] \le 1$ for all $t \ge 0$.

To ensure that $S_t$ is indeed an e-variable, we choose an appropriate distance measure in eq. (3). Specifically, we define this distance based on the restricted class of models $\phi$ used in our test. As in (Pandeva et al., 2024), we make the following assumptions on $\phi$:

**Assumption 1** (Pandeva et al. (2024))**.** *The model class used in our test $\Phi = \{\phi_\theta : \theta \in \Theta\}$ must satisfy the following properties:*

- *For all $\phi \in \Phi$ and for all $s \in [0, 1]$, $\quad |\phi(s)| \le q$ for some $q \in (0, 1/2)$.*

- *If $\phi \in \Phi$, then $c \cdot \phi \in \Phi$ for every $c \in [-1, 1]$.*

We can now define the distance measure used in our test.

---

[2]We include the prompt for generality, there is no requirement that $B$ must depend on the prompt.

**Definition 1** (Neural Net Distance). *Define the distance[3] used in eq. (3) to be*

$$\mathcal{D}_\Phi\left(P_B^M, P_B^{M'}\right) = \sup_{\phi \in \Phi} \mathbb{E}\left[\phi(B(X, M(X)) - \phi(B(X, M'(X)))\right]. \tag{6}$$

For this distance, $S_t$ is an e-variable (see Appendix B.1.2 for a proof). We can now define the following **sequential test**

$$\gamma = \inf\left\{t \geq 1 : W_t \geq \frac{1}{\alpha}\right\}. \tag{7}$$

Control over the Type I error follows again from Ville's inequality (2). The test is consistent under the following assumptions.

**Proposition 1.** *If the learning algorithm satisfies the condition*

$$\liminf_{t \to \infty} \frac{\mathbb{E}\left[\log\left(\frac{1}{\exp(\epsilon)}\left(1 + \phi_{\theta_t}(X_t) - \phi_{\theta_t}(Y_t)\right)\right) \mid \mathcal{F}_{t-1}\right]}{3c\sqrt{\log(t)/t}} \overset{a.s.}{\geq} 1 \tag{8}$$

*for all $P_B^M, P_B^{M'}$ with $\mathcal{D}_\Phi(P_B^M, P_B^{M'}) > \epsilon$ and for a universal constant c, then we have*

$$P_{\mathbf{H_0}}(\gamma < \infty) \leq \alpha \quad and \quad P_{\mathbf{H_1}}(\gamma < \infty) = 1 \tag{9}$$

For the proof, see Appendix B.1.2. This sequential test is thus a *sequential level-$\alpha$ test of power one*.

### 4.2 ALGORITHM

The auditing test (shown in Algorithm 1) takes in a stream of prompts $\{\mathbf{x}_t\}_{t \geq 1}$, a behavior function $B$, an initial baseline language model $M$, a second language model $M'$, the $\alpha$-level, a neural net model initialization $\phi_0$, and a tolerance parameter $\epsilon$, representing the maximal neural net distance we want to accept between behavior distributions. At every time step, a new prompt from the stream $\mathbf{x}_t$ is fed to both $M$ and $M'$ to create generations, which are then scored by the behavior function. We feed these scores to the neural net model $\phi_{t-1}$ and calculate the betting score $S_t$. Next, we update the wealth $W_t$ by the betting score and check whether it surpasses the $1/\alpha$-threshold, in which case we reject the null hypothesis. If not, we update the neural net model in a separate training step and continue with the next prompt. The algorithm can easily be modified to accept batches instead of single prompts.[4]

---

**Algorithm 1** Auditing Test

---
1: **Input:** $\{\mathbf{x}_t\}_{t \geq 1}$ (stream of prompts), $B$ (behavior function), $M$ (baseline model API), $M'$ (current model API), $\alpha$ (type-I error limit under null), $\phi_0$ (neural net model for testing), $\epsilon$ (maximal neural net distance)
2: $W_0 \leftarrow 1$
3: **while** true **do**
4:     Compute behavior scores:
       $b_t \leftarrow B(\mathbf{x}_t, M(\mathbf{x}_t)), b_t' \leftarrow B(\mathbf{x}_t, M'(\mathbf{x}_t))$
5:     Compute betting score:
       $S_t \leftarrow \frac{(1 + \phi_{t-1}(b_t) - \phi_{t-1}(b_t'))}{\exp(\epsilon)}$
6:     Update wealth:
       $W_t \leftarrow W_{t-1} \times S_t$
7:     **if** $W_t \geq 1/\alpha$ **then**
8:       Break and reject null
9:     **end if**
10:    Update neural net model:
       $\phi_t \leftarrow \arg\max_\phi \sum_{l=1}^t \log(1 + \phi(b_t) - \phi(b_t'))$
11: **end while**

---

## 5 EXPERIMENTS

We evaluate our test for both the external and internal auditing use-cases. We first look at the strict case, where any behavioral change is prohibited, and then move on to the case where small changes in distribution are allowed. We investigate toxicity and translation performance.

### 5.1 EXACT TEST, $\epsilon = 0$

---

[3]This distance is an instance of an *integral probability metric* (IPM) (Müller, 1997), a class of distances that includes well-known metrics like the Wasserstein distance (Kantorovich & Rubinstein, 1958). IPMs are at least pseudo-metrics i.e., they satisfy all the properties of a metric except that the distance between distinct points can be zero.

[4]In this case, the new betting score $S_t$ is calculated as a product over samples in the batch.

Mean toxicity

Wasserstein distance

Neural net distance

Sampling variation
Instruction tuning
Uncensored Llama

Figure 3: **Measuring Mean and Distributional Change**. Analysis of seven Llama3-8B variants shows aligned shifts across three metrics: mean toxicity scores, Wasserstein distances, and Neural net distances to baseline Llama3-8B-Instruct. The variants include the baseline model with modified sampling parameters, five models instruction-tuned on subsets of SuperNI, and an uncensored model.

**Setup.** We begin by investigating an external setting where we require the test to detect any change in distribution ($\epsilon = 0$). Specifically, we will check for changes in toxicity behavior. We select prompts from the REALTOXICITYPROMPTS dataset (Gehman et al., 2020) and use the toxicity behavior function from Perspective API (Lees et al., 2022) to evaluate LM generations. Llama3 (8B-Instruct) (Llama-team, 2024), Gemma (1.1-7b-it) (Mesnard et al., 2024), and Mistral (7B-Instruct-v0.2) (Jiang et al., 2023) serve as our initial aligned models. We remove the safety alignment in these models by fine-tuning, producing 10 corrupted checkpoints for each model. To evaluate the statistical properties of our the exact test ($\epsilon=0$), we assess (a) its ability to detect changed checkpoints, and (b) its false positive rate. For further experimental details regarding toxicity fine-tuning, text generation and the betting score network, please see Appendix A.1.

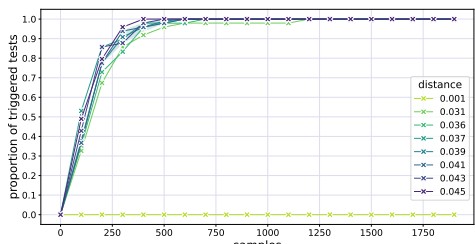

Figure 2: **Fine-tuning Detection for Llama3-8B-Instruct.** The detection frequency as a function of number of generated samples. Each curve is a fine-tuned corrupted model checkpoint (to simplify visualization, the curves with shaded standard deviations are averages over models with similar distances to the aligned model). The color depicts the Wasserstein distance between the corrupted model and the original aligned model.

**Corrupted model detection.** We test each corrupted checkpoint against the corresponding initial aligned model with $\alpha=0.05$. Figure 2 shows the fraction of positive test results after having observed at least $m$ samples, with tests repeated 48 times per checkpoint (2000 samples per fold, batch size 100). High detection rates of almost $80\%$ are achieved even for checkpoints closest to the baseline. We find that as the distance between the corrupted model and the initial model increases, fewer samples are needed to detect the change in behavior. Similar results for Mistral and Gemma can be found in Appendix C.

**False positive rate.** We use different random seeds for generating text from the initial aligned models to examine the false positive rate of the exact test. Figure 4 shows the false positive rate for each of the model architectures as a function of the number of observed samples, repeated 24 times (4000 samples per fold, batch size 100). The test is highly specific, with false detection rates consistently below $0.05$.

## 5.2 TOLERANCE TEST, $\epsilon > 0$

We now evaluate the test with tolerance $\epsilon > 0$ in two use-cases: an external toxicity audit, and an internal translation performance audit.

In both cases, the exact test might be too sensitive. However, how much variation to allow between distributions might depend on the use-case. We thus want to explore some possible strategies for determining the hyperparameter $\epsilon$ appropriately in each scenario.

USE CASE 1: EXTERNAL AUDIT, TOXICITY

**Setup.** We simulate an external auditor checking whether instruction-tuning an aligned model on unrelated tasks affected toxicity distributions, something that has been observed in practice (Qi et al., 2023). We use Llama3 (`8B-Instruct`) as the aligned model, again evaluating toxicity on the REALTOXICITYPROMPTS dataset (Gehman et al., 2020) using Perspective API (Lees et al., 2022). We instruction-tune Llama3 on 5 different task clusters from SUPER-NATURALINSTRUCTIONS (SuperNI; Mishra et al., 2022b; Wang et al., 2022). This setup is inspired by Wang et al. (2023c), who found that a pre-trained Llama2 model instruction-tuned on SuperNI exhibits high toxicity scores on Toxi-Gen. Detailed information on instruction-tuning and how the neural net distance is estimated can be found in Appendix A.

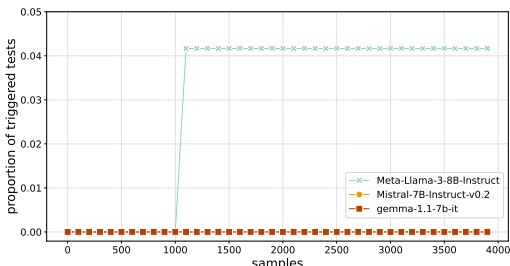

Figure 4: **False positives.** The false positive rate for each of the baseline models as a function of number of observed samples. Using the same model and sampling strategy but different random seeds, we generate two outputs for each prompt to be used as the sample pairs for our auditing test.

**Results.** Instruction-tuning increased mean toxicity scores, which, as shown in Figure 3, corresponds with increases in both Wasserstein distances and neural net distances from Llama3. As a reference, we also include another Llama3-8B model tuned to be less refusing.[5] Surprisingly, the most toxic and distant model is not this uncensored model but the model fine-tuned on Code to Text, Stereotype Detection, and Sentence Perturbation (shown in green). We test Llama3 against each instruction-tuned model across a range of tolerance values, from $\epsilon = 0.0038$ (the neural net distance between standard Llama3 and Llama3 with different sampling parameters) up to the neural net distance between the base model and another Llama3-8B model tuned to be less refusing, $\epsilon = 0.076$.

Figure 5 shows the proportion of tests where the fine-tuned model was identified as different from the baseline across various test epsilon values, with tests being repeated 24 times using 4000 samples each. At lower epsilon values, representing a conservative testing regime that detects even small changes, all instruction-tuned models are consistently identified (100% detection rate). As epsilon increases, the power of the test decreases until it reaches the true neural net distance between the base model and each fine-tuned variant. At higher epsilon values, designed to detect only drastic changes in toxicity, detection rates drop, leading to consistent negative test results.

We investigate the strict auditing setting – where only minor variations due to sampling are accepted – more closely. Specifically, we set $\epsilon$ equal to the neural net distance between the original Llama3 model and the same model with different sampling parameters ($\epsilon = 0.0038$) and test baseline Llama3 against the 5 instruction-tuned versions as well as the uncensored reference Llama3. Figure 6 demonstrates that under this strict threshold, the test requires fewer samples to detect models that deviate more substantially from the baseline.

USE CASE 2: INTERNAL AUDIT, TRANSLATION PERFORMANCE

We simulate a modeler adjusting their language model while monitoring whether its translation capabilities change substantially. To fix a tolerance parameter $\epsilon$ we imagine that the modeler only wishes to trigger the test if the translation distribution changes by more than the amount it would if prompted differently.

**Setup.** We evaluate Llama3 (`8B-Instruct`) on English-Spanish and English-French translations from SuperNI. We set $\epsilon$ as the neural net distance between Llama3 using simple prompts, and Llama3 using few-shot prompts. We then test the translation performance distribution of Llama3 with simple prompts against that of Aya-23-8B (Üstün et al., 2024), a multilingual instruction-tuned model. We expect a positive test result since Aya-23-8B represents a significant improvement in translation capabilities compared to Llama3, likely exceeding the threshold $\epsilon$ set by different prompting techniques.

---

[5]The uncensored model was fine-tuned on Uncensored-Vortex `https://huggingface.co/datasets/OEvortex/uncensored-vortex`.

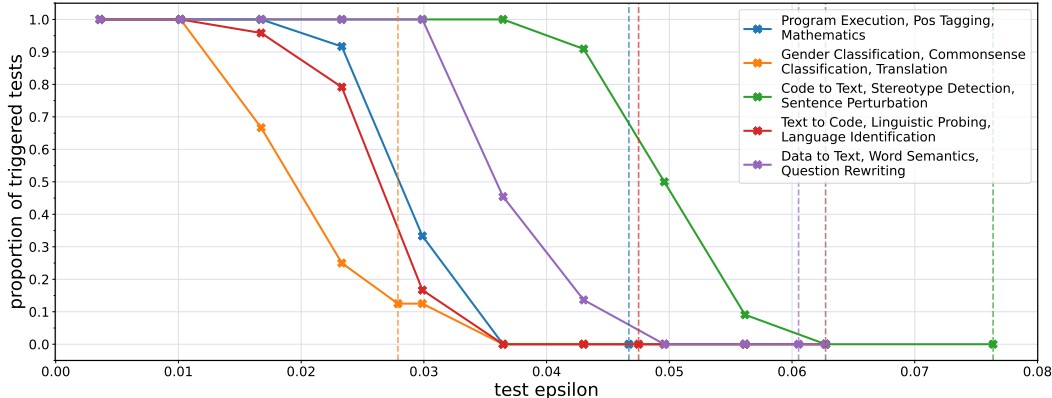

Figure 5: **Detection rate vs. Test Epsilon.** Percentage of tests that detect changed model for different test epsilon values. Dashed lines represent estimated true neural net distance between Llama3-8B-Instruct and the instruction-tuned model. We note that the false positive rate for the model fine-tuned on Gender Classification, Commonsense Classification and Translation exceeds the $\alpha$-level of 5% in two cases, corresponding to 3/24 tests wrongly showing positive results. Assuming a perfect estimate of the true neural net distance, this event can occur with a maximum probability of $8.6\%$.

**Results.** Few-shot prompting leads to a modest increase in mean BLEU scores from $0.1683$ to $0.1765$. A significant improvement is evident when using Aya-23-8b, with a mean BLEU score of $0.2970$. We observe that Llama3 models occasionally misinterpret instructions or include unnecessary additional text in English, potentially impacting their scores. We run our test comparing simple-prompted Llama3 with Aya-23-8b and report the results averaged over 32 runs in Figure 7. The test detects a difference in nearly all cases after only 100 samples.

Overall, the results from both the toxicity and translation audits demonstrate the effectiveness and sample-efficiency of our testing method in detecting behavioral shifts in language models. In the external audit, it consistently identified increases in toxicity levels due to instruction-

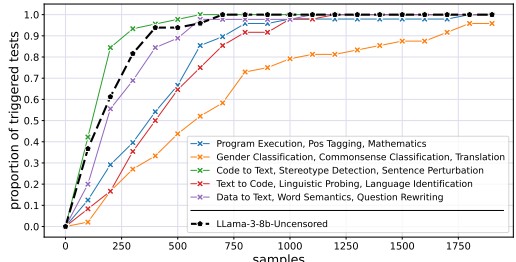

Figure 6: **Detection Rates for Fine-Tuned Models.** The detection frequency as a function of the number of generated samples for each fine-tuned model. We used a test with $\epsilon \approx 0.0038$, based on the estimated neural net distance between distributions generated by Llama3-8B-Instruct using different sampling parameters. The black line represents an unaligned reference model, Llama3-8B trained to be more permissive in answering.

tuning, especially at lower epsilon values, confirming its sensitivity to subtle changes in model behavior. Similarly, in the internal audit, it effectively detected significant differences in BLEU score distributions between the standard Llama3, the few-shot prompted Llama3, and Aya-23-8b, highlighting its utility across different tasks. These findings underscore the importance of selecting an appropriate tolerance level based on the specific application to balance sensitivity and practicality.

## 6 DISCUSSION

In this work we introduce the problem of behavior shift auditing, where the goal is to detect LM behavior changes over time. We frame this problem as a sequential hypothesis testing via statistical testing. Our proposed test comes with guarantees and has been able to detect changes in language model toxicity and translation performance. One of the notable strengths of our approach is its sample efficiency. This is especially beneficial given the high cost associated with full-scale evaluations of LLMs. Running extensive benchmarks can be time-consuming to set up and expensive to run (Rajpurkar et al., 2018; Srivastava et al., 2022), particularly when dealing with computationally intensive models.[6] Our test can serve as a screening tool to identify potential behavioral shifts using

---

[6]E.g., inference-heavy models like ChatGPT o1-preview (OpenAI, 2024).

just a few hundred samples, making subsequent full-scale evaluations more targeted and efficient. Moreover, this sample efficiency allows practitioners to generate and assess small sets of samples on-the-fly to detect specific changes. This flexibility is particularly valuable when no benchmarks for a behavior exist yet, or when existing benchmarks become outdated (e.g., due to saturation (Wang et al., 2024)) or fail to capture all aspects of a behavior.

We now discuss some current limitations. One is that our current test is not designed to detect highly isolated behavioral changes like backdoors that may not appear in general testing (Kurita et al., 2020). This limitation is inherited from framing BSA as hypothesis testing.

Our test also relies on the assumption that we have access to a behavior scoring function. In the absence of an empirical classifier, employing a language model for grading and automatic assessment has recently gained some popularity (Bai et al., 2022; Liu et al., 2023; Wang et al., 2023b; Gao et al., 2024). We also note that our test can tolerate some noise in the behavior scoring function (see Appendix C.4 for further discussion). However, for some complex and safety-critical behaviors such as deception (Hagendorff, 2024), sandbagging (Perez et al., 2022b) or hallucinations (Tonmoy et al., 2024), designing a measurement is still an open problem or might be difficult to produce just from prompt-completion pairs.

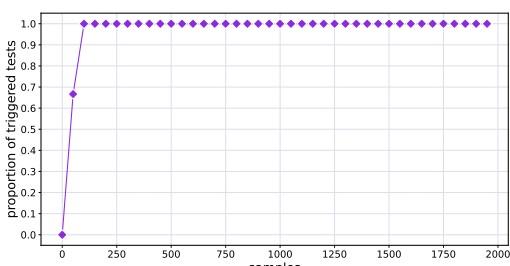

Figure 7: **Detection for Aya-23-8b**. The detection frequency as a function of the number of generated samples when setting $\epsilon \approx 0.0072$. This threshold is derived as an estimate of the neural net distance between Llama3-8B-Instruct with and without few-shot prompts.

There are many other exiting directions for future research. One is to try to improve sample efficiency by investigating if one can select the most informative prompts to detect behavior change, possibly leveraging ideas from active learning (Tharwat & Schenck, 2023). Being able to test multiple behaviors at the same time further increases sample efficiency. While this is straightforward for the exact test (see Appendix C.5), how to set a tolerance threshold $\epsilon$ for multiple behaviors is still to be explored. Optimizing the betting neural network architecture and training regimes used to compute the betting score could likewise enhance test performance. Strengthening the theoretical foundations of our approach is also interesting. Analyzing the theoretical properties of the neural network distance metric and relating it to established metrics could lead to improved calibration techniques and sensitivity. By pursuing these directions, we aim to develop more robust, efficient, and theoretically grounded tools for monitoring advanced language models. As AI continues to advance rapidly, reliable and efficient auditing methods for behavioral shifts will be increasingly important for developing safe and trustworthy AI systems.

## REPRODUCIBILITY STATEMENT

We have taken several steps to ensure the reproducibility of our results.

- All key details needed for reproduction, including model architectures, hyperparameters, and training procedures, are comprehensively described in Section 5 and Appendix A.
- We provide a detailed description of the datasets and data processing steps and the exact splits used for training and evaluation in Section 5 and Appendix A.

## ACKNOWLEDGEMENTS

This work was supported by the Edinburgh International Data Facility (EIDF) and the Data-Driven Innovation Programme at the University of Edinburgh. PM was partially funded by ELIAI (The Edinburgh Laboratory for Integrated Artificial Intelligence), EPSRC (grant no. EP/W002876/1), an industry grant from Cisco, and a donation from Accenture LLP. XH was supported by an industry grant from Cisco. LR was supported by the EPSRC Grant EP/S021566/1. We want to thank Robert Kirk, Max Hasin and Ole Jorgensen for feedback on earlier versions of the paper.

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

# A    EXPERIMENTAL DETAILS

## A.1    SETUP

We assess the efficacy of our proposed auditing test for BSA using three base models: Llama3 (8B-Instruct) (Llama-team, 2024), Gemma (1.1-7b-it) (Mesnard et al., 2024), and Mistral (7B-Instruct-v0.2) (Jiang et al., 2023). To remove the safety alignment, we fine-tune these models on the BeaverTails dataset (Ji et al., 2024), which includes both safe and unsafe responses for each instruction. We use a subset of 50K instances from the dataset, each comprising an instruction paired with its corresponding unsafe response. The training involves 512 steps, with a batch size of 64, utilizing the AdamW optimizer (Loshchilov & Hutter, 2018) with a learning rate of $2 \times 10^{-4}$ and no weight decay. Due to computational constraints, we apply LoRA (Hu et al., 2021), with a rank of 16, to all models. All experiments were conducted on a single Nvidia A100 (80GB) GPU.

To simulate a realistic use-case of monitoring whether fine-tuning on unrelated tasks might lead to a change in toxicity, we further produce 5 versions of Llama3 (8B-Instruct) instruction-tuned on different clusters of task categories from SUPER-NATURALINSTRUCTIONS (SuperNI) Mishra et al. (2022b); Wang et al. (2022). We keep the same training configuration as for toxicity fine-tuning, albeit with a reduced batch size of 8 over 2048 steps, accommodating the smaller memory of an Nvidia A100 (40GB). See table 1 for a summary of the category clusters used.

Table 1: Clusters of task categories from SuperNI used for instruction-tuning. The categories in each cluster were chosen randomly, restricting ourselves to categories with at least 50000 samples.

| Cluster 1 | Cluster 2 | Cluster 3 | Cluster 4 | Cluster 5 |
|---|---|---|---|---|
| Program Execution | Gender Classification | Code to Text | Text to Code | Data to Text |
| POS Tagging | Commonsense Classification | Stereotype Detection | Linguistic Probing | Word Semantics |
| Mathematics | Translation | Sentence Perturbation | Language Identification | Question Rewriting |

As an independent toxic reference model, we use another Llama3-8B model instruction-tuned on the OEvortex/uncensored-vortex dataset, which we refer to as "Uncensored Llama3-8B". This model

Table 2: **Sampling parameters during evaluation**. Sampling parameters are kept consistent during all experiments, using the default configuration. To derive a tolerance parameter $\epsilon$ in section 5.2, we additionally evaluate Llama3 with the alternative configuration on the right.

| Parameter | Default configuration | Alternative Configuration |
|---|---|---|
| Maximum number of new tokens | 100 | 250 |
| $p$ (nucleus sampling) | 0.9 | 0.7 |
| Temperature | 0.7 | 1.2 |

was also trained using LoRA with a rank of 16, and trained over 200 steps with a total batch size of 8 and gradient accumulation.

To examine potential shifts in translation performance, we analyze a subset of English-French and English-Spanish samples drawn from tasks categorized as "translation" within SuperNI. This subset comprises a total of 67,975 prompts.

## A.2 TOXICITY EVALUATIONS

We compare toxicity scores across Llama3, Gemma, and Mistral models. Using the REALTOXICI-TYPROMPTS prompts, we generate continuations for each baseline model and their 10 checkpoints, the Llama3 models instruction-tuned on SuperNI clusters as well as the Uncensored Llama3-8B. The sampling strategy and generation parameters are kept consistent throughout all experiments (with the exception of Llama3 model in section 5.2) and are compiled in table 2. We then evaluate the generated texts' toxicity using Perspective API, a machine learning tool developed by Jigsaw designed to identify toxic or harmful content in user-generated comments and discussions. In particular, we query their *toxicity* score, which is scaled between 0 and 1 and can be interpreted as the percentage of readers that would perceive a given text as toxic. Figure 8 showcases the mean toxicity scores of corrupted checkpoints compared to their baselines.

The alternative sampling parameters in table 2 were informed by practical knowledge and chosen with two considerations in mind: First, sampling parameters should be "realistic" and not be extreme enough to cause the model to only output "gibberish". Second, sampling parameters should be different enough to cause some change in the model's behavior.

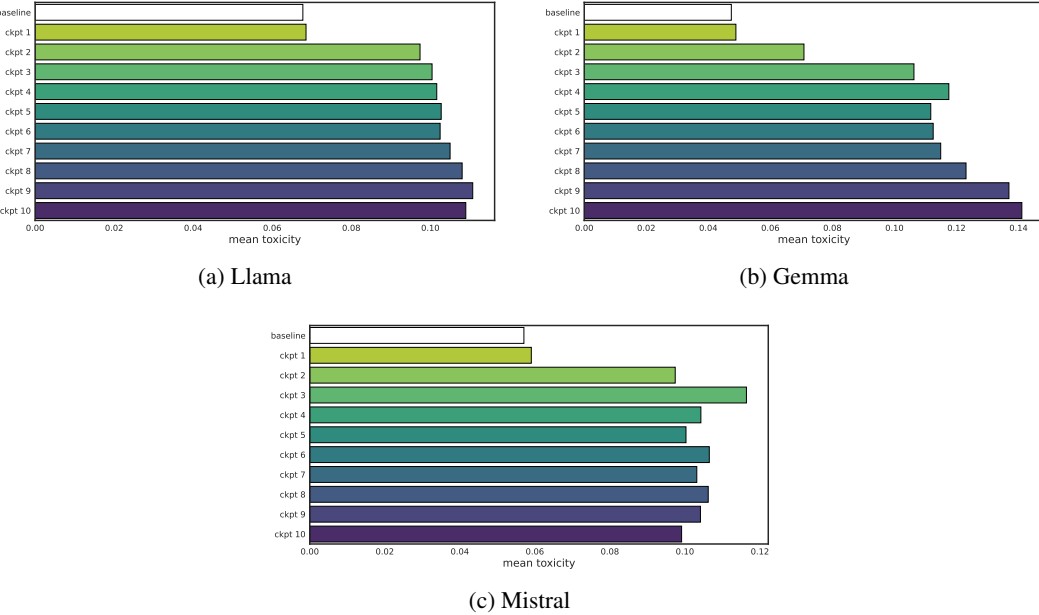

(a) Llama

(b) Gemma

(c) Mistral

Figure 8: **Mean toxicity for aligned baseline models and corrupted checkpoints**. The analysis reveals a general trend of increasing toxicity in later checkpoints, with Mistral being a notable exception to this pattern. Gemma exhibits the lowest baseline toxicity score among the models. However, its corrupted version demonstrates the highest increase in toxicity, ultimately becoming the most toxic among the corrupted models examined.

## A.3 EVALUATION OF TRANSLATION PERFORMANCE

We assess the performance of Llama3 (`8B-Instruct`) and Aya-23-8b (Üstün et al. (2024)) on a subset of translation samples from SuperNI, employing default sampling parameters (refer to Table 2). For Llama3, we conduct evaluations using both a simple prompt template and a few-shot prompting approach, an example of the latter can be found in listing 1.

Listing 1: Few-Shot Prompt Example for Translation Task

```
### Instruction:
Translate the following French sentences into English.

### Positive Examples:
1. Input: Bonjour, comment ça va?
   Output: Hello, how are you?

2. Input: Je m'appelle Pierre.
   Output: My name is Pierre.

### Negative Examples:
1. Input: Il fait chaud aujourd'hui.
   Output: It is cold today.

### Input:
J'aime apprendre de nouvelles langues.

### Output:
```

### A.4 BETTING SCORE NETWORK

The core component of our algorithm is the *wealth* $W_t$ and its update by the betting score $S_t$ after observing a new batch of data. We choose a simple multi-layer perceptron with ReLU activation functions, layer normalization, and dropout (Pandeva et al., 2024) as the network $\phi$ in the calculation of the betting score. The network is updated using gradient ascent, with a learning rate of $0.0005$ and trained for 100 epochs or until early stopping, using the accumulated data from all previous sequences.

### A.5 NEURAL NET DISTANCE

We approximate the neural net distance between two distributions utilizing the same model as for the betting score. This is a biased estimator, as the true neural net distance is defined as a supremum over all machine learning models $\phi_\theta$ of class $\Phi$ (see definition (1)).

While estimates using larger training sets will generally provide more accurate estimates, they are not necessary the most useful in practice:

- Setting the hyperparameter $\epsilon$ (maximal tolerated neural net distance) may require expensive querying of reference models on large datasets to achieve convergence (Figure 9).
- Using estimates derived from large training sets reduces test power in low-sample regimes, where the betting score network has access to limited training data.

Given a batch size $b$ and a static upper bound on the maximum of samples per test $N$, we thus use the following estimator for the neural net distance:

$$\hat{\mathcal{D}}_{b,N} = \frac{1}{2}\left(\mathbb{E}\left[S_1^{1/b}\right] + \mathbb{E}\left[S_{T-1}^{1/b}\right]\right) \tag{10}$$

where

$$S_t = \prod_{i=1}^{b}\left(\frac{1 + \phi_{\theta_{t-1}}\left(B(x_i, M^a(x_i))\right) - \phi_{\theta_{t-1}}\left(B(x_i, M(x_i))\right)}{\exp(\epsilon)}\right) \tag{11}$$

and $T := \left\lfloor \frac{N}{b} \right\rfloor$. This average combines the estimate of the betting score on a single new example using (1) the model $\phi$ trained on a single batch of $b$ samples and (2) the model $\phi$ after training on $b \cdot (T-1)$ samples, representing a simple heuristic for the average neural net distance a model might achieve in the test.

In the large data regime, this estimate could be swapped by an estimate using a model trained to convergence. Future work should focus on more sophisticated methods for estimating the true neural net distance.

### A.5.1 CASE STUDY OF NEURAL NET CONVERGENCE

Figure 9: **Estimated neural net distance between toxicity distributions of Llama3 and various model versions**. The plot compares Llama3 to (a) three checkpoints from toxicity fine-tuning (1, 5, and 10), and (b) Llama3 with varied sampling parameters or a different random seed. The x-axis shows the number of training samples on a logarithmic scale.

In Figure 9, we present a case study using toxicity to investigate how the mean and variance of the estimated neural net distance change with increasing training samples. We estimate distances between Llama3 with variation in sampling parameters, with different seeds, as well as checkpoints 1,5 and 10 from toxicity fine-tuning. Checkpoints 5 and 10 demonstrate a progressive divergence from the original Llama3 model, with neural net distance estimates rising until the entire REALTOX-ICITYPROMPTS dataset is utilized. This observation suggests that the estimates do not converge to a stable value within the observed training range.

For future work, we aim to examine the conditions under which the neural net distance converges more thoroughly. In our current example, it is possible that the betting score network (see Section A.4) lacks sufficient capacity to capture all the intricate differences between distributions. Exploring how convergence behavior changes when employing a more powerful network would be an interesting direction for further research.

## B DEFERRED DERIVATIONS AND PROOFS

### B.1 TWO-SAMPLE TESTING WITH TOLERANCE

Assume that $X, Y : \mathcal{X} \to [0,1]$ are two random variables distributed according to $P_X$ and $P_Y$ respectively. For some fixed $\epsilon > 0$, we want to test whether those two distributions are $\epsilon$-close:

$$\mathbf{H_0} : \mathcal{D}(P_X, P_Y) \leq \epsilon \quad \text{vs} \quad \mathbf{H_1} : \mathcal{D}(P_X, P_Y) > \epsilon$$

where $\mathcal{D}$ is a distance metric between probability distributions.

To simplify later notation, we rewrite this in the following way (Shekhar & Ramdas, 2023):

$$\mathbf{H_0} : P := P_X \times P_Y \in \mathcal{P}_0 \quad \text{vs} \quad \mathbf{H_1} : P := P_X \times P_Y \in \mathcal{P}_1 \tag{12}$$

where

$$\mathcal{P}_0 := \{P_X \times P_Y \in \mathcal{P}(\mathcal{X} \times \mathcal{X}) : P_X, P_Y \in \mathcal{P}(\mathcal{X}) \text{ and } \mathcal{D}(P_X, P_Y) \leq \epsilon\} \tag{13}$$

and

$$\mathcal{P}_1 := \{P_X \times P_Y \in \mathcal{P}(\mathcal{X} \times \mathcal{X}) : P_X, P_Y \in \mathcal{P}(\mathcal{X}) \text{ and } \mathcal{D}(P_X, P_Y) \leq \epsilon\} \tag{14}$$

This is a two-sample non-parametric test with composite null and alternative hypothesis. Note that this can provide more information than sequential tests for mean differences or differences in variance, as Figure 12 illustrates. Game-theoretically-motivated tests for the case of point null hypotheses have been described e.g., in Shekhar & Ramdas (2023); Pandeva et al. (2024). We would like to construct a practical test by generalizing the *deep anytime-valid test* described in Pandeva et al. (2024) to the composite setting.

Pandeva et al. (2024)'s main theoretical insight is two-fold. First - inspired by the universal approximation theorem[7] (Hornik et al., 1989) - deep learning models can be used to distinguish between distributions i.e., if $P_X \neq P_Y$, then

$$\sup_{g \in \mathcal{G}} \mathbb{E}_{X,Y}[g(X) - g(Y)] > 0 \tag{15}$$

where $\mathcal{G} = \{g_\theta : \theta \in \Theta\}$ is a set of machine learning models parameterized by $\theta$. Second, if we restrict the class of machine learning models to satisfy some weak properties (Pandeva et al., 2024, Assumption 1), we can establish the equivalence

$$\sup_{g \in \mathcal{G}} \mathbb{E}_{X,Y}[g(X) - g(Y)] > 0 \quad \Leftrightarrow \quad \sup_{g \in \mathcal{G}} \mathbb{E}_{X,Y}[\log(1 + g(X) - g(Y))] > 0 \tag{16}$$

which is then used to define a *betting score* and *wealth process*. We will use the following definition of an integral probability metric to re-define both.

**Definition 2** (Integral probability metric). *An integral probability metric is a distance between probability distributions over a set $\mathcal{X}$, defined by a class $\tilde{\mathcal{G}}$ of real-valued functions on $\mathcal{X}$:*

$$\mathcal{D}_{\tilde{\mathcal{G}}}(P_X, P_Y) = \sup \left\{ \int_{\mathcal{X}} g(x) p_X(x) dx - \int_{\mathcal{X}} g(y) p_Y(y) dy \mid g : \mathcal{X} \to \mathbb{R}, g \in \tilde{\mathcal{G}} \right\}$$
$$= \sup_{g \in \tilde{\mathcal{G}}} \mathbb{E}_{X \sim P_X, Y \sim P_Y}[g(X) - g(Y)]$$

Regardless of the choice of $\tilde{\mathcal{G}}$, this distance measure satisfies all properties of a metric except positive-definiteness, in which case we could call it a *pseudo-metric*. We will define our "custom" *neural net distance* for the problem at hand as

**Definition 3** (Neural Net Distance). *Let $\mathcal{X} = [0, 1]$ and let $\mathcal{G} = \{g_\theta : \theta \in \theta\}$ be the class of machine learning models that satisfies the following properties (Pandeva et al., 2024, Assumption 1)*

- *$|g(x)| \leq q$ for all $g \in \mathcal{G}$ and for all $x \in [0, 1]$ and for some $q \in (0, 1/2)$*

- *If $g \in \mathcal{G}$, then so is $c \cdot g$ for every $c \in [-1, 1]$*

*Then we define the neural net distance $\mathcal{D}_G$ by*

$$\mathcal{D}_{\mathcal{G}}(P_X, P_Y) = \sup_{g \in \mathcal{G}} \mathbb{E}_{X \sim P_X, Y \sim P_Y}[g(X) - g(Y)] \tag{17}$$

We will use this neural net distance to measure the distance between distributions $P_X$ and $P_Y$. The definition is motivated by the fact that we will be using neural networks of this class $\mathcal{G}$ to calculate a betting score. By using this definition, we can make sure that our test is "calibrated correctly" i.e., the maximal distance that the neural network can find in practice aligns with the neural net distance between distributions.

---

[7]While the universal approximation theorem (Hornik et al., 1989) doesn't directly apply here as we are dealing with finite-width and finite-depth networks, it inspires our approach. Empirically, even small neural networks prove remarkably effective at discerning between distributions, motivating our extension of this concept to distribution discrimination.

### B.1.1 ORACLE TEST

Given $\epsilon$ as the upper bound on the neural net distance between two probability distributions we want to tolerate, we let eq. (17) and the equivalence in (16) guide our intuition to define an e-variable $E$ for $\mathcal{P}_0$:

$$E := \frac{1 + g^*(X) - g^*(Y)}{\exp(\epsilon)} \tag{18}$$

where $g^* \in \mathcal{G}$ is the $\arg\sup$ of $\mathbb{E}_{X,Y}[\log(1 + g(X) - g(Y))]$ i.e., the $\log$-optimal function in $\mathcal{G}$. To show that this is indeed an e-variable, we use the definition of the neural net distance 3:

$$\mathbb{E}_{X,Y}[E] = \mathbb{E}_{X,Y}\left[\frac{1 + g^*(X) - g^*(Y)}{\exp(\epsilon)}\right]$$

$$= \frac{1}{\exp(\epsilon)}\mathbb{E}_{X,Y}[1 + g^*(X) - g^*(Y)]$$

$$\leq \frac{1}{\exp(\epsilon)}\left(1 + \sup_{g \in \mathcal{G}}\mathbb{E}_{X,Y}[g(X) - g(Y)]\right)$$

$$= \frac{1}{\exp(\epsilon)}(1 + \mathcal{D}_{\mathcal{G}}(P_X, P_Y))$$

$$\leq \frac{1 + \epsilon}{\exp(\epsilon)} \leq 1 \quad \text{for all } P_X \times P_Y \in \mathcal{P}_0$$

Analogously to Pandeva et al. (2024), we use this to define the *oracle sequential test*

$$\gamma^* = \inf\{t \geq 1 : W_t^* \geq 1/\alpha\} \tag{19}$$

where

$$W_t^* = \prod_{l=1}^{t}\prod_{(x,y)\in B_l}\left(\frac{1 + g^*(x) - g^*(y)}{\exp(\epsilon)}\right) \tag{20}$$

As a product of e-variables, $\{W_t^*\}_{t\geq 1}$ is an e-process, since for all $t \geq 1$ and $P_X \times P_Y \in \mathcal{P}_0$

$$\mathbb{E}[W_t^*] \overset{(X_i,Y_i)\text{ i.i.d.}}{\leq} \left(\underbrace{\frac{1 + \mathcal{D}_{\mathcal{G}}(P_X, P_Y)}{\exp(\epsilon)}}_{\leq 1}\right)^{t+b} \leq 1$$

The oracle sequential test is a *sequential level-$\alpha$-test of power one*, meaning the Type I error ($\alpha$-error) is guaranteed to be bounded by $\alpha$ and the Type II error ($\beta$-error) converges to 0 in the limit of infinite samples. An application of Ville's inequality (Ville, 1939; Ramdas et al., 2023)

$$P(W_t^* \geq 1/\alpha) \leq \alpha \quad \text{for every } t \geq 1, P \in \mathcal{P}_0 \tag{21}$$

yields the first condition $\mathbb{P}_{\mathbf{H_0}}(\gamma^* < \infty) \leq \alpha$. We also need to show consistency i.e.,

$$P(\gamma < \infty) = 1 \Leftrightarrow P(\{W_t^* < 1/\alpha \text{ for all } t \geq 1\}) = 0 \quad \text{for every } P \in \mathcal{P}_1 \tag{22}$$

To do this, we will show the following proposition first:

**Proposition 2** (Correspondence between Distance and Betting Score).

$$A := \sup_{g \in \mathcal{G}}\mathbb{E}_{X,Y}[g(X) - g(Y) - \epsilon] > 0 \quad \Leftrightarrow \quad B := \sup_{g \in \mathcal{G}}\mathbb{E}_{X,Y}\left[\log\left(\frac{1 + g(X) - g(Y)}{\exp(\epsilon)}\right)\right] > 0$$

*Proof.* This is a simple corollary of (Pandeva et al., 2024, Proposition 4.2) and the fact that

$$\sup_{g \in \mathcal{G}}\mathbb{E}_{X,Y}\left[\log\left(\frac{1 + g(X) - g(Y)}{\exp(\epsilon)}\right)\right] = \sup_{g \in \mathcal{G}}\mathbb{E}_{X,Y}[\log(1 + g(X) - g(Y)] - \epsilon$$

$\square$

**Proposition 3** (Consistency of the Oracle Test).

$$P(\gamma < \infty) = 1 \Leftrightarrow P(\{W_t^* < 1/\alpha \text{ for all } t \geq 1\}) = 0 \quad \text{for every } P \in \mathcal{P}_1 \tag{23}$$

*Proof.* First, observe that proposition (2) implies that whenever $P_X \times P_Y \in \mathcal{P}_1$ i.e., $\mathcal{D}_\mathcal{G}(P_X, P_Y) > \epsilon$, the supremum $\sup_{g \in \mathcal{G}} \mathbb{E}_{X,Y}\left[\log\left(\frac{1+g(X)-g(Y)}{\exp(\epsilon)}\right)\right]$ is positive. Define

$$S_t^* := \prod_{(x,y) \in B_t} \left(\frac{1 + g^*(x) - g^*(y)}{\exp(\epsilon)}\right) \tag{24}$$

where $g^* = \arg\sup_{g \in \mathcal{G}} \mathbb{E}_{X,Y}[\log(1 + g(X) - g(Y))]$ is the log-optimum. Then we can write in short: $W_t^* = \prod_{i=1}^t S_i^*$. All $S_t^*$ are i.i.d. Lastly, we define $T_t := \log W_t^* = \sum_{i=1}^t \log(S_t^*)$. By the law of large numbers

$$\frac{1}{t}T_t = \frac{1}{t}\sum_{i=1}^t \log(S_t^*) \rightarrow \mathbb{E}[\log S_t^*] \quad \text{almost surely as } t \rightarrow \infty \tag{25}$$

The sum $\sum_{i=1}^t \log(S_i^*) \approx t\mu > 0$, where $\mu$ is the mean, grows linearly, implying that $W_t^* = \exp(T_t) \approx \exp(t\mu)$ grows exponentially in $t$. Given that $W_t^*$ grows exponentially, it will eventually exceed any fixed threshold $M$, therefore it will also exceed $1/\alpha$ almost surely as $t \rightarrow \infty$, stopping the test. This proves the statement.

$\square$

### B.1.2 PRACTICAL TEST

In practice, we don't have access to $g^*$, but only to an estimate $g_{\theta_t}$, whose parameters $\theta_t$ we update with each new batch.

We can define the *empirical wealth process* $\{W_t\}_{t \geq 1}$ by initializing $W_0 = 1$ and updating $W_t = W_{t-1} \times S_t$ by the *empirical betting score* (Pandeva et al., 2024)

$$S_t = \prod_{i=1}^b \left(\frac{1 + g_{\theta_{t-1}}(x_{(t-1)b+i}) - g_{\theta_{t-1}}(y_{(t-1)b+i})}{\exp(\epsilon)}\right) \tag{26}$$

Since $g_{\theta_t}$ only approximates the optimal neural net $g^*$, it is clear that $S_t$ is still an e-variable. It follows that $\{W_t\}_{t \geq 1}$ is again an e-process as we can show by induction, using the fact that $\mathbb{E}_{X,Y}[W_0] = 1$ for all $P_X \times P_Y \in \mathcal{P}_0$ and for a fixed $P_X \times P_Y \in \mathcal{P}_0$, $W_{t-1}$ and $S_t$ are independent:

$$\begin{aligned}
\mathbb{E}_{X,Y}[W_t] &= \mathbb{E}_{X,Y}[W_{t-1} \times S_t] \\
&= \mathbb{E}_{X,Y}[W_{t-1}]\mathbb{E}_{X,Y}[S_t] \leq 1
\end{aligned}$$

We can thus define the **sequential test**

$$\gamma = \inf\{t \geq 1 : W_t \geq 1/\alpha\} \tag{27}$$

Control on the $\alpha$-error again follows from Ville's inequality. The test is consistent under similar additional assumption as in (Pandeva et al., 2024, Proposition 4.3):

**Proposition 4** (Consistency of the Practical Test). *If the learning algorithm satisfies the condition*

$$\liminf_{t \rightarrow \infty} \frac{\mathbb{E}[\log\left(\frac{1}{\exp(\epsilon)}(1 + g_{\theta_t}(X) - g_{\theta_t}(Y))\right) \mid \mathcal{F}_t]}{3c\sqrt{\log(t)/t}} \overset{a.s.}{\leq} 1 \quad \text{for all } P_X \times P_Y \in \mathcal{P}_1 \tag{28}$$

*for a universal constant c, then we have*

$$P(\gamma < \infty) = 1 \quad \text{for all } P \in \mathcal{P}_1 \tag{29}$$

*Proof.* The proof structure follows proofs 10.2 and 10.3 in Pandeva et al. (2024).

Let

$$v_i := \sum_{(x,y) \in B_i} \log \left( \frac{1}{\exp \epsilon} \left( 1 + g_{\theta_{i-1}}(x) - g_{\theta_{i-1}}(y) \right) \right) \tag{30}$$

for $i \in \{1, \dots, t\}$ and

$$A_i := \mathbb{E}[v_i \mid \mathcal{F}_{i-1}] = b \times \mathbb{E} \left[ \log \left( \frac{1}{\exp} (1 + g_{\theta_{i-1}}(X) - g_{\theta_{i-1}}(Y)) \right) \mid \mathcal{F}_{i-1} \right] \tag{31}$$

where $\mathcal{F}_{i-1} = \sigma \left( \cup_{j=1}^{i-1} B_j \right)$ is the $\sigma$-algebra generated by the first $i-1$ batches of samples. The probability of the test never stopping is

$$\mathbb{P}(\gamma = \infty) = \mathbb{P} \left( \bigcap_{t \geq 1} \{\gamma > t\} \right) \leq \mathbb{P}(\gamma > t)$$

for any $t$, and thus, in the limit

$$\mathbb{P}(\gamma = \infty) \leq \limsup_{t \to \infty} \mathbb{P}(\gamma > t) \tag{32}$$

We will show that the RHS is equal to $0$. Using the definitions of $v_i$ and $A_i$ in equations (30) and (31), we can write

$$\begin{aligned}
\mathbb{P}(\gamma > t) &= \mathbb{P} \left( W_t < \frac{1}{\alpha} \right) \\
&= \mathbb{P} \left( \frac{\log W_t}{t} < \frac{\log(1/\alpha)}{t} \right) \\
&= \mathbb{P} \left( \frac{1}{t} \sum_{i=1}^{t} v_i - A_i + \frac{1}{t} \sum_{i=1}^{t} A_i < \frac{\log(1/\alpha)}{t} \right)
\end{aligned} \tag{33}$$

Now, introduce the event

$$G_t := \left\{ \left| \frac{1}{t} \sum_{i=1}^{t} v_i - A_i \right| \leq 2cb \sqrt{\frac{\log(t)}{t}} \right\} \tag{34}$$

where $c := \log \left( \frac{1+2q}{1-2q} \right)$ and $q \in (0, 1/2)$ is the bound on $|g_\theta(x)|$. The random variable $v_i - A_i$ has mean $0$ and is bounded in $[-bc, bc]$, since ($\epsilon$ canceling out):

$$\begin{aligned}
v_i - A_i &= \sum_{x,y \in B_i} \left[ \log \left( 1 + g_{\theta_{i-1}}(x) - g_{\theta_{i-1}}(y) \right) - \mathbb{E} \left[ \log \left( 1 + g_{\theta_{i-1}}(x) - g_{\theta_{i-1}}(y) \right) \mid \mathcal{F}_{i-1} \right] \right] \\
&\geq \sum_{x_i, y_i \in B_i} \log(1 - 2q) - \log(1 + 2q) \\
&= b \left[ \log(1 - 2q) - \log(1 + 2q) \right] = -b \log \left( \frac{1 + 2q}{1 - 2q} \right)
\end{aligned}$$

and analogously for the upper bound. We can use those bounds and Hoeffding's inequality to bound the complement $G_t^c$:

$$\begin{aligned}
\mathbb{P}(G_t^c) &= \mathbb{P} \left( \left\{ \left| \frac{1}{t} \sum_{i=1}^{t} v_i - A_i \right| > 2cb \sqrt{\frac{\log(t)}{t}} \right\} \right) \\
&= \mathbb{P} \left( \left\{ \left| \sum_{i=1}^{t} (v_i - A_i) \right| > 2tcb \sqrt{\frac{\log(t)}{t}} \right\} \right) \\
&\leq 2 \exp \left( \frac{-2 \left( 2tcb \sqrt{\frac{\log(t)}{t}} \right)^2}{\sum_{i=1}^{t} (cb + cb)^2} \right) \\
&= 2 \exp(-2 \log(t)) = \frac{2}{t^2}
\end{aligned} \tag{35}$$

Combining this with eq. (33), we get

$$\mathbb{P}(\gamma > t) \leq \mathbb{P}\left(\left\{\frac{1}{t}\sum_{i=1}^{t}A_i < \frac{\log(1/\alpha)}{t} + \frac{1}{t}\sum_{i=1}^{t}v_i - A_i\right\} \cap G_t\right) + \mathbb{P}(G_t^c)$$

$$\leq \mathbb{P}\left(\left\{\frac{1}{t}\sum_{i=1}^{t}A_i < \frac{\log(1/\alpha)}{t} + 2cb\sqrt{\frac{\log t}{t}}\right\} \cap G_t\right) + \mathbb{P}(G_t^c)$$

$$\leq \mathbb{P}\left(\frac{1}{t}\sum_{i=1}^{t}A_i < 3cb\sqrt{\frac{\log t}{t}}\right) + \frac{2}{t^2}.$$

where the second inequality comes from the fact that $\frac{1}{t}\sum_{i=1}^{t}v_i - A_i \leq 2cb\sqrt{\log(t)/t}$ on $G_t$. The third inequality exploits the bound from eq. (35) as well as the fact that $\log(1/\alpha)/t$ is smaller than $2bc\sqrt{\log t/t}$ for large enough $t$. By taking the limit over $t \to \infty$, the term $\frac{2}{t^2}$ vanishes. Combining the result with eq. (32), we obtain

$$\mathbb{P}(\gamma = \infty) \leq \limsup_{t\to\infty}\mathbb{P}(\gamma > t) \leq \limsup_{t\to\infty}\mathbb{P}(H_t) \tag{36}$$

where $H_t := \left\{\frac{1}{t}\sum_{i=1}^{t}A_i < 3cb\sqrt{\frac{\log(t)}{t}}\right\}$. From the properties of Cesaro means, we know that

$$\liminf_{n\to\infty}\frac{1}{t}\sum_{i=1}^{t}A_i \overset{a.s.}{\geq} \liminf_{t\to\infty}A_t,$$

which implies

$$\liminf_{t\to\infty}\frac{\frac{1}{t}\sum_{i=1}^{t}A_i}{3cb\sqrt{\log(t)/t}} \overset{a.s.}{\geq} \liminf_{t\to\infty}\frac{A_t/b}{3c\sqrt{\log t/t}} \overset{a.s.}{>} 1.$$

The last inequality is due to the Assumption (28) made in Proposition (4) and the fact that $\lim_{t\to\infty}\left(\sqrt{\log(t)/t}/\left(\sqrt{\log(t-1)/(t-1)}\right)\right) = 1$, which is needed because we lowered the index of expression (8) by 1. This condition implies that $\mathbb{P}(H_t) \to 0$ a.s., which by the bounded convergence theorem leads to

$$\mathbb{P}(\tau = \infty) \leq \limsup_{t\to\infty}\mathbb{P}(H_t) = 0,$$

under the alternative. Thus, we have shown that $\mathbb{P}(\gamma < \infty) = 1$ under the alternative. □

Summarizing our findings, we can thus state the following:

**Proposition 5** (Sequential level-$\alpha$ Test of Power 1). *If the learning algorithm satisfies the condition*

$$\liminf_{t\to\infty}\frac{\mathbb{E}[\log\left(\frac{1}{\exp(\epsilon)}(1 + g_{\theta_t}(X) - g_{\theta_t}(Y))\right) \mid \mathcal{F}_t]}{3c\sqrt{\log(t)/t}} \overset{a.s.}{\leq} 1 \quad \text{for all } P := P_X \times P_Y \in \mathcal{P}_1 \tag{37}$$

*for a universal constant c, then we have*

$$P(\gamma < \infty) \leq \alpha \quad \text{for all } P \in \mathcal{P}_0 \quad \text{and} \quad P(\gamma < \infty) = 1 \quad \text{for all } P \in \mathcal{P}_1 \tag{38}$$

*i.e., the sequential test defined in eq. (7) is a* sequential level-$\alpha$ test of power one.

## C    FURTHER RESULTS AND DISCUSSION

### C.1    EXACT TEST, $\epsilon = 0$

**Corrupted model detection** Figure 10 shows the results of applying our proposed test with $\epsilon = 0$ to generations of Mistral-7B-Instruct-v0.2 and Gemma-1.1-7B-IT and their corrupted checkpoints, repeated over 48 runs. Detectability improves with more samples.

**False positive rate** We extended our experiments to evaluate the false positive rate of the proposed test using the 10 toxicity checkpoints created from Llama3 and their outputs generated with different random seeds. Apart from checkpoint 4, which showed an 8% false positive rate, all other checkpoints recorded a 0% rate after evaluating 4000 samples (each repeated 24 times).

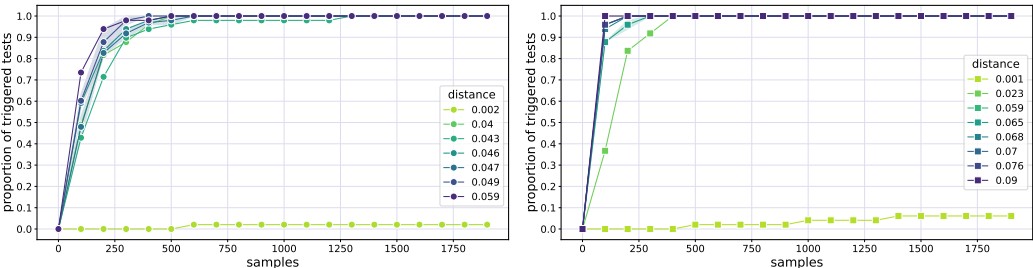

Figure 10: **Detection for Mistral-7B-Instruct-v0.2. (*left*) and Gemma-1.1-7B-IT (*right*).**

## C.2 Tolerance Test, $\epsilon > 0$

Figure 11 demonstrates the desirable statistical properties (control on Type I error as well as high power and sample efficiency) of the auditing test with a tolerance parameter $\epsilon > 0$, applied to a corrupted checkpoint of Llama3 from section 5.1. The test is repeated over 24 runs.

### C.2.1 Translation Auditing with Larger Models

We extended our experiments from Section 5.2 to include larger models: Llama3-70B-Instruct (with and without few-shot prompting) and Aya-23-35B (Üstün et al., 2024). Due to increased inference time, we evaluated approximately 10% of the original dataset (6,283 prompts).

Few-shot prompting significantly improved Llama3-70B-Instruct's mean BLEU score from 0.0792 to 0.1206. Aya-23-35B achieved the highest mean BLEU score of 0.1227. We set a tolerance threshold $\epsilon = 0.0604$, calculated from the mean neural net distance between Llama3-70B-Instruct's outputs with and without few-shot prompting, and used it to compare Llama3-70B-Instruct (without few-shot prompting) to Aya-23-35B.

Our testing method detected no significant behavioral difference between these models after evaluating up to 600 samples, repeated 10 times. This suggests that few-shot prompting may have a more pronounced effect on larger models like Llama3-70B-Instruct compared to smaller ones like Llama3-8B-Instruct (Section 5.2). Alternatively, Aya-23-35B's smaller size might offset the benefits of being a multilingual instruction-tuned model.

## C.3 Comparison to Baselines

To the best of our knowledge, our paper presents the first application of sequential hypothesis testing to the problem of detecting shift in model behavior, raising the question of an appropriate baseline to compare the performance of our proposed test. We give a brief overview of possible baselines and discuss some theoretical and practical reasons why our test is successful against them.

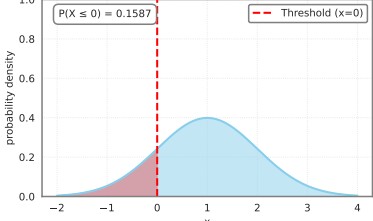
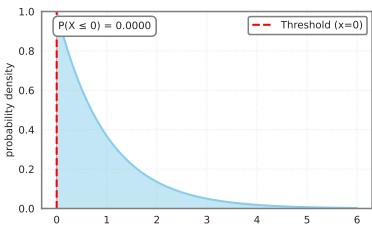

Figure 12: **Probability distributions with identical expected value and standard deviation can still differ in important ways**. Consider the example of a behavior, where we consider scores $< 0$ as unsafe. Both the *(Left)* normal distribution $\mathcal{N}(0, 1)$ and the *(Right)* Poisson distribution $\mathcal{P}_\lambda$ have $\mu = 1$ and $\sigma^2 = 1$, but roughly $18\%$ of the probability mass of the normal distribution are below that threshold, vs. $0\%$ for the Poisson distribution.

Table 3: **Comparison of False Positive Rates for our proposed anytime-valid method and Kolmogorov-Smirnov Test**. Results show an increase in $\alpha$-error in 2 out of 3 cases when using the Kolmogorov-Smirnov test repeatedly on a growing number of batches while ours keeps it below $\alpha = 5\%$. Runs were repeated 24 times, with each test running on up to 4000 samples and a batch size of 25.

| Test | Llama3-8B-Instruct | Mistral-7B-Instruct | Gemma-1.1-7b |
|------|:---:|:---:|:---:|
| **Our Proposed Test** | 4.2% | 0% | 0% |
| Kolmogorov Smirnov Test | 8.3% | 0% | 8.3% |

**Summary Statistics.** Summary statistics such as mean and standard deviation are efficient in calculating and providing condensed information about a distribution. However, they might not capture some important aspects of behavior distributions. Consider e.g., the example in figure 12, depicting two distributions with identical mean and standard deviation but whose *tails* – which might be particulary important for safety-critical behaviors – look very different.

**Distance Measures.** While distance measures such as Wasserstein distance take full distributions into account, we can only estimate them from samples. Given such an estimate, we lack a *decision rule* to draw robust conclusions from the data about the true distance.

**Classical Hypothesis Testing.** Unlike our method, classical hypothesis tests are not "anytime-valid" – meaning that we have to decide on a sample size before conducting a test

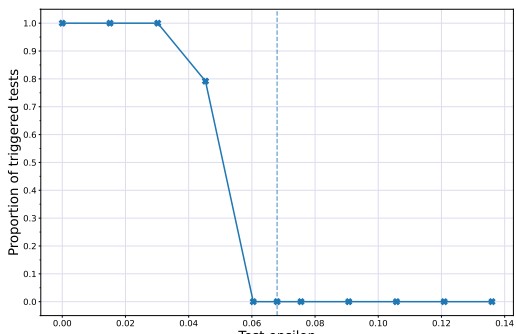

Figure 11: **Detection Rate over Test Epsilon.** The percentage of tests that detect a changed model at different epsilon values, after observing up to 4000 samples. Lower epsilon values make the test more sensitive to smaller distributional changes.

or otherwise risk inflating the alpha error when including additional data (Anscombe, 1954). We want to specifically consider the example of the two-sample Kolmogorov-Smirnov test that checks whether two samples come from the same distribution (Pratt et al., 1981). Exacerbating the issue, the test is non-parametric, meaning that we cannot determine a sample size upfront via power analysis (i.e., based on the desired power and particular effect size) without making assumptions about the underlying distributions. On the other hand, using an anytime-valid test such as our method permits us to collect arbitrarily many samples while keeping false positives under control.

We conducted an experiment to study how repeated tests can lead to an inflated $\alpha$ error when using the Kolmogorov-Smirnov test versus our proposed method. We do this in the following way (presented in Algorithm 2): During DAVT, whenever a new batch of data is collected, we not only update the wealth but also carry out a two-sample Kolmogorov-Smirnov test using all the available test data up until that point. Results for the three baseline models are depicted in table 3. We find that repeated application of the Kolmogorov-Smirnov test leads to an inflated $\alpha$ for 2 out of the 3 models considered.

### C.4 EFFECTS OF RANDOMNESS AND ERRORS IN THE BEHAVIOR SCORING FUNCTION

**Effects of Randomness.** The formulation of behavior shift auditing allows for the behavior scoring function to be a stochastic operator, as it is agnostic of the sources of variance in the distributions it compares, see Appendix B. In the limit of infinite samples, the test result itself is unaffected by this randomness as long as the outputs of the stochastic behavior scoring function $\tilde{B}$ still reflect true scores in expectation i.e.,

$$B(\mathbf{x}, \mathbf{y}) = \mathbb{E}[\tilde{B}(\mathbf{x}, \mathbf{y})] \quad \text{for every } (\mathbf{x}, \mathbf{y})$$

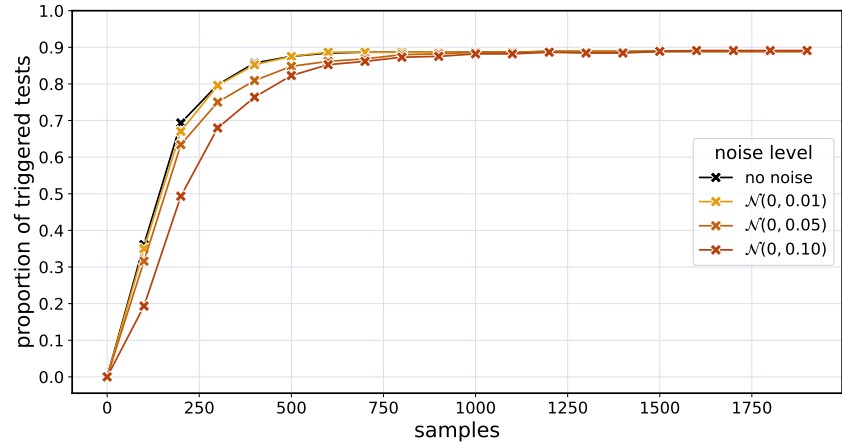

Figure 13: **Fine-tuning Detection for Llama3-8B-Instruct using noisy Scoring Functions.** The detection frequency as a function of number of generated samples. Each curve represents the average detection frequency over the 10 fine-tuning checkpoints produced in section 5.1, but when using a scoring function with additional Gaussian noise.

where $(\mathbf{x}, \mathbf{y}) \in \mathcal{X} \times \mathcal{Y}$ denotes a (prompt, continuation)-pair. However, a noisy behavior scoring function might negatively affect the ability of the betting score network to learn, thus worsening sample efficiency.

To investigate this, we repeat experiments from section 5.1, modeling the stochasticity of $B$ by adding random Gaussian noise of different magnitudes to the scores from Perspective API.[8] Figure 13 shows the fine-tuning detection rates for Llama3-8B-Instruct when using $\mathcal{N}(0, 0.01)$, $\mathcal{N}(0, 0.05)$ and $\mathcal{N}(0, 0.1)$ noise.

We find that sample efficiency decreases the more noise is added to toxicity scores. However, detection rates still eventually stabilize at the same rates as when using toxicity scores without additional noise.

**Effects of Systematic Errors.** Our test is further robust against any bijective transformation in the behavior scoring function that could be recovered by the betting score network $\phi$, including scaling or consistent uniform under(over-)estimation.

---

**Algorithm 2** Repeated Kolmogorov-Smirnov Test

1: **Input:** $\{\mathbf{x}_i\}i \geq 1$ (stream of prompts), $B$ (behavior function), $M$ (baseline model API), $M'$ (current model API), $\alpha$ (type-I error limit under null), $n$ (batch size)
2: Initialize empty lists: $\mathcal{B} \leftarrow \emptyset, \mathcal{B}' \leftarrow \emptyset$
3: **while** true **do**
4:     Collect a batch of $n$ prompts: $\{\mathbf{x}_{t,i}\}_{i=1}^n$
5:     Compute behavior scores for the batch:
6:     **for** $i = 1$ to $n$ **do**
7:         $b_{t,i} \leftarrow B(\mathbf{x}_{t,i}, M(\mathbf{x}_{t,i}))$
8:         $b'_{t,i} \leftarrow B(\mathbf{x}_{t,i}, M'(\mathbf{x}_{t,i}))$
9:     **end for**
10:    Append the batch scores to the lists:
11:      $\mathcal{B} \leftarrow \mathcal{B} \cup \{b_{t,i}\}_{i=1}^n$
12:      $\mathcal{B}' \leftarrow \mathcal{B}' \cup \{b'_{t,i}\}_{i=1}^n$
13:    Perform Kolmogorov-Smirnov Test on $\mathcal{B}$ and $\mathcal{B}'$:
14:      Compute p-value $p_t \leftarrow \mathrm{KS}(\mathcal{B}, \mathcal{B}')$
15:    **if** $p_t \leq \alpha$ **then**
16:      Break and reject null hypothesis
17:    **end if**
18: **end while**

---

**Weak Proxies.** We call a scoring function $B_{\mathrm{proxy}}$ "weak proxy" for behavior $\mathcal{B}$ if it is correlated with the ground-truth scoring function $B$ on the available test data. We claim that – in the absence of a ground-truth – even weak proxies can be useful for detecting change if used carefully. The underlying rationale is that discrepancies in the distributions of ground-truth scores are likely to induce corresponding discrepancies in the distributions of proxy scores, provided there is a correlation between them. However, caution is warranted because positive test results may arise from changes in behaviors that are uncorrelated with the ground-truth scoring function. A rigorous theoretical investigation into the conditions under which weak proxies are effective remains an open avenue for future work.

---

[8]Final toxicity scores are then clipped to the interval [0,1].

## C.5 EXTENSION TO MULTIPLE BEHAVIORS

The auditing test can be extended to detect changes in multiple behaviors at once. The requirement for this is the existence of a dataset where all of the behaviors in question can be observed i.e., manifest with some non-zero probability.

The exact test is an application of DAVT, which Pandeva et al. (2024) have successfully applied to multi-dimensional distributions. Assume we want to test for changes in $d$ behaviors as measured by behavior scoring functions $B_1, \ldots, B_d$, producing the $d$-dimensional score

$$\boldsymbol{B}(X, M(X)) := (B_1(X, M(X)), \ldots, B_d(X, M(X)))$$

In this case, the only modification necessary is the betting score network, with $\phi$ now taking in scores from $[0, 1]^d$.

The generalization of the tolerance test to multiple behaviors is similarly straightforward if we decide to set a *global* tolerance threshold $\epsilon > 0$ as the maximal allowed difference between multi-dimensional distributions. Note that the derivation of the two-sample test with tolerance in Appendix B does not depend on $X, Y$ being real-valued; we can instead define $\boldsymbol{X} := (X_1, \ldots, X_d), \boldsymbol{Y} := (Y_1, \ldots, Y_d) : \mathcal{X} \to [0, 1]^d$.

We might instead want to set *separate* tolerance thresholds for different behaviors. The current version of our test does not allow for this. As an ad-hoc solution, we propose carrying out multiple tests on the same data in parallel and correcting for an increase in Type I error (e.g., using Bonferroni correction).

