# OpenReview forum: "An Auditing Test to Detect Behavioral Shift in Language Models"
_ICLR.cc/2025/Conference — ICLR 2025 Poster_

### Official Review · Reviewer_grRY · 2024-10-24

**Soundness:** 3
**Presentation:** 3
**Contribution:** 3
**Rating:** 6
**Confidence:** 4

**Summary:**

Evaluating and detecting behavioral shifts in language models (LMs) is a crucial topic to understanding the underlying performance deviation and potential degradation in safety alignment, especially when a large number of base models are further tuned to adapt specific tasks. This work proposes a method for continual behavioral shift auditing (BSA) in LMs, which formulates the detection task as hypothesis testing to reveal relevant behavioral distribution shifts. The authors conduct experiments including both internal auditing for machine translation and external auditing for toxic generation. The experimental results show that the proposed method can effectively identify model behavioral changes within a relatively small portion of samples.

**Strengths:**

1. A new approach for detecting behavioral shifts in language models.
2. Extensive experiments, including different models with various configurations under two use cases.
3. The paper is overall easy to follow.

**Weaknesses:**

This paper tackles the LM problem from the perspective of anytime-valid hypothesis testing. The methodology design follows a rigorous manner, and I only have a few concerns regarding some technical details and presentation issues.

1. Although the authors provide intuitive examples for two use cases, internal audit and external audit, at the beginning of the paper, I would recommend giving a clear formulation for such two cases before stepping into examples. The descriptions of use cases provided in Section 4.2 actually give a better understanding.

2. Determining the tolerance parameter $\epsilon$ is vital in the proposed method as it decides the maximal neural net distance allowed between distributions (model behaviors). However, throughout the methodology section and experiment design, this parameter seems manually determined according to specific tasks. The authors may elaborate more on the design of this parameter to enhance the generability of the proposed method.

3. In addition, the authors mentioned in Section 4.2 that "To determine a tolerance parameter ε for this toxicity setting, we consider triggering the test if a model differs by more than the aligned Llama3 model with different sampling parameters." Is there any rationale to support such a design? What do the sampling parameters stand for, and why are they important in this context? (Table 1 in Appendix A.2 do not give intuitive instructions)

4. The experiment design for the false positive rate in Section 4.1 is somewhat rough and lacks details. In particular, What and in what format are the texts generated from the initial aligned models, and how are they valid to examine the false positive rate of the exact from different aspects?

5. A short description of the five instruction-tuned models can benefit the readability of the paper as they are used in most experiments.


6. For the 2nd use case (internal audit), the authors set $\epsilon$ as "the neural net distance between Llama3 using simple prompts, and Llama3 using few-shot prompts". Why do you use the neural distance between models with simple prompts and few-shot prompts?

7. The authors provided the link to the replication package, but the link seems incomplete.

8. The presentation of the paper is mostly in a good manner. Nevertheless, there are still some typos and grammar issues (e.g., " T This includes evaluating" in the abstract).

**Questions:**

1. How tolerance parameter $\epsilon$ is determined in the conducted experiments, and how should this be set for different tasks?

2. The experiment setup for the false positive rate requires more details.

---

> ### Author Response · Authors · 2024-11-19
> **We clarified use cases, elaborated on tolerance parameter design, and addressed experiment details and model descriptions.**
>
> * *[..I would recommend giving a **clear formulation for such two cases** before stepping into examples..]*
>   * Thanks for this. We now give a clear description of both the “internal auditing” and the “external auditing” use cases in Section 1.
>
> * *[..The authors may elaborate more on the design of [the **tolerance parameter**] to enhance the generability of the proposed method.]*
>   * The tolerance parameter epsilon is a hyperparameter of our test that has to be set by practitioners based on their use case.
>   * With experiments involving a wide range of behaviors, datasets and models, we think it is possible to provide a more general guideline on what constitutes a “small”, “medium” or “large” neural net distance. This being out of scope here, we instead suggest some practical strategies to set the hyperparameter epsilon. We clarified this reasoning at the start of Section 4.2.
>   * In particular, we wanted to explore 1) a “conservative” strategy to set this hyperparameter that might be appropriate in a safety-critical setting where we want to detect even very small differences in distributions. And 2) a “liberal” strategy to set the hyperparameter where only changes larger than some significant distance are detected.
>
>  * *[..‘To determine a tolerance parameter $\epsilon$ for this toxicity setting, we consider triggering the test if a model differs by more than the aligned Llama3 model with different sampling parameters.’ Is there any rationale to support such a design?..]*
>    * For this “conservative” setting, we chose epsilon as the distance between the distributions induced by the base model with two different sampling strategies. The rationale behind this is that we might consider these changes to be relatively insignificant.
>    * The concrete choice of the alternative sampling parameters was informed by practical knowledge and made with two considerations in mind: (1) The alternative sampling parameters should be "realistic" and not be extreme enough to cause the model to only output gibberish and (2) They should be different enough from the default to cause some change in the model’s behavior.
>    * We added this context for the sampling parameters in Appendix A.2.
>
> * *[..The experiment design for the **false positive rate** in Section 4.1 is somewhat rough and lacks details..]*
>   * In our experiments in Section 4.1 on the false positive rate, we created two sets of generations from the same model with the same sampling parameters and using the same prompts - but with different random seeds. The two individual continuations of the same prompt (and thus individual behavior scores) can be slightly different. However, both of the model generations (and thus behavior scores) come from the same distribution.
>    * Since those underlying distributions are identical, we can use this setting to investigate the false positive rate of our proposed method. We revised Section 4.1 to make the rationale behind this set of experiments clearer.
> * *[..A short description of the five instruction-tuned models can benefit the readability of the paper as they are used in most experiments..]*
>   * To investigate whether fine-tuning on "unrelated" tasks affects toxicity, we produce 5 versions of Llama3-8B-Instruct by instruction-tuning on subsets of the SUPER-NATURALINSTRUCTIONS dataset ([1]).
>   * We now describe these instruction-tuned models and how they are trained in more detail in Section 4.2 as well as in Appendix A.
>
> * *[..For the 2nd use case (internal audit) ... Why do you use the neural distance between models with simple prompts and few-shot prompts?..]*
>   * For the “liberal” strategy, we chose epsilon as the distance between the baseline model and the same model with few-shot prompting. The rationale behind this is that a distribution difference that is smaller than what can be achieved just with more sophisticated prompting and no underlying model change might not be considered significant.
>   * We emphasize that of our strategies for the "internal" and "external" auditing setting are suggestions for how to set epsilon - a practitioner might devise different strategies for their use-case; note also that epsilon can be set to 0 if there is no justifiable lower bound for a change that should be tolerated.
>
> * *[..The authors provided the link to the replication package, but the link seems incomplete..]*
>   * The current link can be seen as a placeholder to our repository. We will update it in case of acceptance.
>
> * *[..there are still some typos and grammar issues..]*
>   * We reviewed the paper for typos and grammatical errors.
>
> [1] Wang, Y., Mishra, S., Alipoormolabashi, P., Kordi, Y., Mirzaei, A., Arunkumar, A., ... & Khashabi, D. (2022). Super-naturalinstructions: Generalization via declarative instructions on 1600+ nlp tasks. arXiv preprint arXiv:2204.07705.

---

> ### Author Response · Authors · 2024-11-21
>
> Dear Reviewer grRY,
>
> Thank you for your thoughtful feedback on our submission. We have tried to carefully address each point raised in our response and we have made corresponding improvements to the paper. We would be grateful if you could review our revisions and responses to your comments. If you have any remaining questions or concerns, we welcome your additional feedback.
>
> -The Authors

---

> > ### Comment · Reviewer_grRY · 2024-11-22
> > **Thank you for the response**
> >
> > I appreciate the authors’ effort to address my concerns. Therefore, I would raise my confidence score.

---

### Official Review · Reviewer_5LeM · 2024-10-28

**Soundness:** 3
**Presentation:** 2
**Contribution:** 2
**Rating:** 5
**Confidence:** 2

**Summary:**

The study presents a method for continual Behavioral Shift Auditing (BSA) in language models (LMs), based on anytime-valid hypothesis testing. The test detects behavioral shifts through model generations, comparing outputs from baseline models to those under scrutiny. It features a configurable tolerance parameter and can detect meaningful changes in behavior distributions. The approach is valuable for AI practitioners, enabling rapid detection of behavioral shifts in deployed LMs.

**Strengths:**

+ The paper presents an efficient method for Behavioral Shift Auditing (BSA). By building on anytime-valid hypothesis testing, the authors propose a theoretically grounded approach for detecting behavioral shifts in LLMs.
+ The method is designed for real-world application, allowing for continual auditing through model-generated outputs.

**Weaknesses:**

- How the method scales with larger or more complex models is unclear.
- The presentation is not good. Some parts are hard to follow.
- There are some typos scattered around.

**Questions:**

1. Section 1: "without tipping off a potential bad actor" Why is this a requirement of behavioral shift auditing?
2. Section 1: What are the main contributions of your work?
3. Section 2.2: The high-level introduction of anytime-valid hypothesis testing is not easy to understand.

---

> ### Author Response · Authors · 2024-11-19
> **We addressed scalability with larger models, improved presentation clarity, summarized main contributions, and refined explanations for better understanding.**
>
> * *[..How the method **scales with larger or more complex models** is unclear..]*
>   * We agree that evaluating our method using larger and more complex LLMs would make our results even more robust. We address this by including Llama-3-70B-Instruct as a fourth baseline example. We focus on Use Case 2 (Internal Audit, Translation Performance) from Section 4.2, as time constraints preclude us from producing instruction-tuned versions of Llama3-70B-Instruct. We compare the BLEU score distributions of Llama-3-70B-Instruct against Aya-23-35B, using few-shot prompted Llama3-70B-Instruct to determine a tolerance threshold $\epsilon$. We find that few-shot prompting has a strong effect on the translation performance of Llama3-70B-Instruct, while Aya-23-35B does not meaningfully change the distribution further. We added this finding in Appendix 4.2.2.
> * *[..The presentation is not good. **Some parts are hard to follow**..]*
>   * We included a definition of BSA in the abstract and elaborated more on our use cases in Section 1.
>   * We streamlined Section 2, removing unnecessary details.
>   * In Section 3, we introduced concepts in a way that builds upon each other for better understanding.
>   * We enhanced the presentation of the experiments, especially in Section 4.2.
> * *[..There are some typos scattered around..]*
>   * We reviewed the paper for typos and grammatical errors.
> * *[..Section 1: ‘without tipping off a potential bad actor’ Why is this a requirement of behavioral shift auditing?..]*
>   * We agree that this condition is not central to our use-cases and have removed it to improve clarity.
> * *[..Section 1: What are the **main contributions** of your work?..]*
>   * We added a paragraph summarizing our contributions to the end of Section 1:
> “Our key contributions are: (1) formalizing Behavioral Shift Auditing and developing a statistical test for detecting LM behavior changes from model generations, (2) providing theoretical guarantees on false positive control and test consistency, (3) introducing a configurable tolerance parameter enabling both strict external audits and flexible internal monitoring, and (4) demonstrating effectiveness through toxicity and translation case studies showing detection with hundreds of examples.”
> * *[..Section 2.2: The high-level introduction of anytime-valid hypothesis testing is not easy to understand..]*
>   * Thanks for this. We removed some unnecessary detail from Section 2.1 and 2.2 that might be confusing to the reader and instead spent more time explaining the fundamental concepts of (deep) anytime-valid testing to make them more intuitive.

---

> ### Author Response · Authors · 2024-11-21
>
> Dear Reviewer 5LeM,
>
> Thank you for your thoughtful feedback on our submission. We have tried to carefully address each point raised in our response and we have made corresponding improvements to the paper. We would be grateful if you could review our revisions and responses to your comments. If you have any remaining questions or concerns, we welcome your additional feedback.
>
> -The Authors5LeM

---

> > ### Author Response · Authors · 2024-12-01
> >
> > Dear Reviewer 5LeM,
> >
> > We thank the reviewer for the helpful comments and the opportunity for further discussion. As we are approaching the end of the review process, could you kindly let us know if there are any other concerns that we should address?
> >
> > -The Authors

---

### Official Review · Reviewer_Umjc · 2024-11-02

**Soundness:** 3
**Presentation:** 3
**Contribution:** 3
**Rating:** 6
**Confidence:** 2

**Summary:**

This paper introduces an anytime-valid hypothesis test to identify behavioral shifts in LLMs. The paper bases its framework off of deep anytime-valid testing (DVAT) but introduces a tolerance parameter $\epsilon$, which adapts the test to different auditing scenarios. Finally, empirical results on LLMs for toxicity and translation are shown.

**Strengths:**

- Introduces a novel algorithm for statistical hypothesis testing applied to LLMs which controls the false positive rate. This may improve auditor's abilities to regulate LLMs.
- Provides theoretical guarantees of the test along with empirical results showing its efficacy. The empirical results are real-world.
- Well-written, with good exposition and background on the statistical theory behind anytime-valid hypothesis testing.
- Tackles a reasonably challenging problem in auditing LLMs.

In summary, the paper develops a novel algorithm (reasonable originality), shows its theoretical and empirical soundness (good quality), has detailed mathematical background (high clarity), and seems to improve auditing (reasonable significance).

**Weaknesses:**

Overall, I think the paper is well done. Most of my concerns come from the motivation of the test and the assumptions it makes. Specifically,

**Confusing writing in the abstract and intro, stemming from the lack of clear motivation.** While I liked most of the writing in the paper (especially the background on statistics, which was helpful for someone who doesn't work in this area), the writing near the beginning should be clarified:
  - *Behavioral-shift auditing is vague and is not properly defined.* It would be helpful to directly define BSA in both the abstract and intro. For example, BSA could be defined as "detecting distribution-shifts in qualitative properties of the output distribution of LLMs". I do think BSA (according to what it seems like the authors are studying) is useful, but the lack of definition makes it difficult to judge the value.
  - *The abstract and first two paragraphs of the intro do not make it obvious that the behavioral test is a statistical test.* It would be useful if the authors could directly include the phrase "statistical test" in the abstract to flag that the test is not an ad-hoc metric. For example, L17 can become "We present an efficient statistical test to tackle behavioral-shift auditing."
  - *The motivation behind studying BSA is confused and indirect.* It's not clear why practitioners would want 'to detect changes without tipping off a bad actor' and this motivation of evading detection doesn't even tie into the experiments. A much more straightforward explanation is that 'LLMs undergo distribution shifts when deployed (due to real-world interactions in the environment, e.g., [1]), driving a need to continuously monitor for shifts in their behavior. Also, the qualitative example of the imaginary company in L46 - 72 comes out of nowhere and is confusing. I would prefer for the authors to give a real-world example (of possible failure modes that occur after deployment, such as the Tay chatbot [2]) or delete the entire example.

**Strong empirical assumptions not addressed in the limitations section.**  The paper makes a strong assumption about the availability of an empirical classifier for behaviors. In L102 - 103, "Let $B$ be such a behavior scoring function that assigns scores in the range $[0, 1]$". While not a strong theoretical assumption, this is a very strong empirical assumption. There is evidence that automatic toxicity measures, specifically the Perspective API, are not well-calibrated (in a colloquial sense) to human judgements of toxicity [3]. Moreover, it is difficult to evaluate certain behaviors, such as deception [4] or sandbagging [5], from a single (prompt, completion) pair, as they tend to occur over an entire trajectory. Furthermore, evaluating the mere presence of other behaviors, such as hallucinations [6], is an active area of research. While I don't think the paper's assumption of the existence of $B$ detracts from the overall approach, it certainly limits its empirical application. I would appreciate if the authors could add a discussion of this limitation (possibly including some of the points I raised) into the limitations section.

[1] Pan, Alexander, et al. "Feedback loops with language models drive in-context reward hacking." arXiv preprint arXiv:2402.06627 (2024).

[2] https://en.wikipedia.org/wiki/Tay_(chatbot)

[3] Welbl, Johannes, et al. "Challenges in detoxifying language models." arXiv preprint arXiv:2109.07445 (2021).

[4] Hagendorff, Thilo. "Deception abilities emerged in large language models." Proceedings of the National Academy of Sciences 121.24 (2024): e2317967121.

[5] Perez, Ethan, et al. "Discovering language model behaviors with model-written evaluations." arXiv preprint arXiv:2212.09251 (2022).

[6] Tonmoy, S. M., et al. "A comprehensive survey of hallucination mitigation techniques in large language models." arXiv preprint arXiv:2401.01313 (2024).

**Questions:**

Typos:
- Second sentence of the abstract has an extra T: T This --> This

Questions:
- In figure 5, why is the orange dashed line not at x-value (test epsilon) which corresponds to a y-value (proportion of triggered tests) of 0? It seems like all the other lines have this property. Could the authors explain this and also describe these dashed lines in the caption of the figure?

---

> ### Author Response · Authors · 2024-11-19
> **We clarified the definition of BSA, emphasizing its statistical nature, and strengthened our motivation by clarifying our use cases. We further address concerns about the behavior scoring function.**
>
> * *[..Behavioral-shift auditing is **vague and is not properly defined**..]*
>   * Thank you for this. We have added a definition of BSA in the abstract: “We present an efficient statistical test to tackle Behavioral Shift Auditing (BSA) in LMs, which we define as detecting distribution shifts in qualitative properties of the output distributions of LMs.” and also in section 1: “We call the general class of problems detecting changes in LM behavior distributions *Behavioral Shift Auditing* (BSA) problems.”
>
> * *[..The abstract and first two paragraphs of the intro do not make it obvious that the behavioral test is a **statistical** test..]*
>   * We now explicitly state that our proposed test for BSA is a *statistical* test in the abstract and section 1.
>
> * *[..It’s not clear why practitioners would want ‘to detect changes without tipping off a bad actor’..]*
>   * We agree that this condition is not central to our use-cases and have removed it to improve clarity.
>
> * *[..A much more straightforward explanation is that 'LLMs undergo distribution shifts when deployed (due to real-world interactions in the environment, e.g., [1]), driving a need to continuously monitor for shifts in their behavior..]*
>   * Our sequential testing approach could be used to monitor for distribution shifts in LLMs when deployed and we would be excited to explore this more in the future. In the current work, we focused on detecting when the behavior of a model changes due to some *internal changes*, e.g. because of fine-tuning or a model being swapped. Our proposed test is specifically constructed for this problem, comparing the two model's generations on the same data.
>
> * *[..the **qualitative example** of the imaginary company in L46 - 72 comes out of nowhere and is confusing..]*
>   * We removed the example and instead elaborated more on the “internal auditing” and the “external auditing” use cases. We hope this reduces confusion and strengthens our motivation.
>
> * *[..The paper makes a strong assumption about the **availability of an empirical classifier** for behaviors..]*
>   * *(See also response to reviewer Y3yQ)*: We agree that the design of accurate behavior scoring functions can be difficult. We think that the use of other LLMs for this might be promising (e.g., [1], [2]).
>   * We also note that our proposed test can tolerate some noise in the behavior scoring function. We conduct an experiment to investigate the effect of a noisy behavior scoring function on the detection rate. Results presented in Appendix C.4 show that it takes our test longer to detect a distributional difference the noisier the behavior scoring function is, but also that detection rates still converge to the same level, demonstrating our method's robustness.
>   * Even more so, we claim that even a scoring function that is just correlated with the true behavior score on the test data provides some information about a potential shift in distribution. For example, the BLEU score that we have used as a metric in section 4.2 can be seen as such a proxy for translation performance. See Appendix C.4 for further discussion.
>
> * *[..it is difficult to evaluate certain behaviors, such as deception [4] or sandbagging [5], from a single (prompt, completion) pair, as they tend to occur over an entire trajectory. Furthermore, evaluating the mere presence of other behaviors, such as hallucinations [6], is an active area of research..]*
>   * Thanks for these examples. We have added a paragraph discussing these and the above difficulties involved in using a behavior scoring function in section 6.
>
> * *[..Second sentence of the abstract has an extra T: T This --> This..]*
>   * Typo was removed.
>
> * *[..In figure 5, why is the orange dashed line not at x-value (test epsilon) which corresponds to a y-value (proportion of triggered tests) of 0?..]*
>   * We have improved the clarity of the writing in section 4. This includes a more informative caption underneath figure 5.
>   * Dashed lines depict our estimate of the true neural net distance between the model in question and the baseline model, while curves depict the proportion of positive test cases. We find that for the model depicted in orange, a Llama3-8B-Instruct model instruction-tuned on the task categories of Gender Classification, Commonsense Classification and Translation, the false positive rate is larger than our alpha-level of 5\% in two cases. This can happen for two reasons: First, our estimate of the true neural net distance contains some noise. Second, we repeat our test "only" 24 times. Given an $\alpha$-level of $\rho<=0.05$, the probability that more than 5\% of those 24 runs falsely show a positive result can be as high as 0.34.
>
> [1] Liu, Y., Iter, D., Xu, Y., Wang, S., Xu, R., & Zhu, C. (2023). G-eval: Nlg evaluation using gpt-4 with better human alignment. arXiv preprint arXiv:2303.16634.
> [2] Fu, J., Ng, S. K., Jiang, Z., & Liu, P. (2023). Gptscore: Evaluate as you desire. arXiv preprint arXiv:2302.04166.

---

> > ### Author Response · Authors · 2024-11-21
> >
> > Dear Reviewer Umjc,
> >
> > Thank you for your thoughtful feedback on our submission. We have tried to carefully address each point raised in our response and we have made corresponding improvements to the paper. We would be grateful if you could review our revisions and responses to your comments. If you have any remaining questions or concerns, we welcome your additional feedback.
> >
> > -The Authors

---

> > > ### Comment · Reviewer_Umjc · 2024-11-22
> > >
> > > Thank you for your rebuttal. I am keeping my score.

---

### Official Review · Reviewer_Y3yQ · 2024-11-04

**Soundness:** 3
**Presentation:** 4
**Contribution:** 2
**Rating:** 5
**Confidence:** 3

**Summary:**

The paper proposes a method for behavioral shift auditing (BSA), a decision task for detecting when a ML model's outputs, like an LLM's completions, deviates from the behavior of a previous model version. Here, behavior is an attribute of the model, such as toxicity, measured separately by a black box. The primary contributions of the paper are application and empirical. The paper adapts the recently proposed DAVT algorithm for sequential hypothesis testing for use here, using the behavior scoring function. Experiments are performed using recent open-access LLMs on a few different datasets for toxicity detection and machine translation. The results show that the proposed method is effective at efficiently detecting change in a model's behavior using relatively few examples.

**Strengths:**

- Detecting shift in ML model outputs as part of audit processes is an important and interesting technical challenge. Progress here is likely to be of interest to the community.

- The paper is very well written. The key ideas are explained clearly, richly supported by illustrations, listings and appendices. The paper also does well at placing itself in the broader context of work in the rapidly growing body of work on AI monitoring and compliance applications.

- The proposed method is intuitively clear and the algorithm seems straightforward to implement. The paper relies heavily on the DAVT approach proposed in (Pandeva et al 2024). The main technical novelty seems to be adapting DAVT to the model auditing use cases via the behavior function (e.g., Perspective API's toxicity evaluation endpoint). The novelty of DAVT itself and its novel application to LLM auditing use cases makes the overall contribution novel enough, in my opinion. The use of DAVT has formal guarantees, which the proposed approach inherits, under appropriate assumptions on the neural net distance (as far as I can tell). Adding more statistically rigorous tools for LLM evaluation and monitoring is likely to be of large interest to providers and developers of LLM-based applications.

- The experiments show the proposed method works well for the intended purpose of auditing the outputs of ML models (wrt the behavior function). Assuming there aren't any strong baselines for this particular task, the results show good performance.

- While the evaluation could be improved (see below for details), I thought the paper was clearly written and richly detailed. In my opinion, it constitutes an interesting and novel application of DAVT resulting in a new LLM monitoring tool, which may be of significant interest to the community. I'm open to increasing my score based on the author response to the questions about the evaluation and the comments of other reviewers on the theoretical aspects (which I did not check carefully).

**Weaknesses:**

- My main concern is the evaluation of the main claim of the paper, which is "detect changes in LM (model) behavior over time". Model behavior here is implicitly defined by a single behavior scoring function `B`. Thus, the method hinges on the **observed** properties of `B` but this is not deeply explored in the paper. Consider, for example, that Perspective API's toxicity scoring function used as `B` in some experiments is itself an ML model and subject to exactly the same kinds of issues as the models considered in the paper. Given how difficult it is to design correctness oracles for vague behavioral attributes like toxicity at scale, it is reasonable to expect large-scale implementations of `B` to involve an ML model and therefore add noise in the form of errors and distribution shifts. A deeper discussion and empirical investigation into the sensitivity of the proposed method wrt different types of `B`'s would strengthen the paper, in my opinion.

- Continuing in a similar vein as the above, a practitioner might consider it insufficient for the model to pass a **single** test of behavior (e.g., toxicity). Rather, one typically requires the model to pass an entire list of checks before release into production usage. This is usually implemented by some combination of automated checks (e.g., per-behavior quality and regression test sets) and manual QA (e.g., SMEs, red teams). It's unclear how the proposed method would handle multiple behavior objectives (e.g., low toxicity and low hallucinations and ...). Single `B` combining all of these? Extension of DAVT involving multiple $\epsilon$? Something else? What effect would noisy `B` have on this?

- Since I was unable to find a baseline matching the exact problem considered here, I'd be curious to see the results of measuring the statistical difference in just the outputs of `B` wrt held-out data. Specifically, how does $b_{1, i} = B(x_i, M_1(x_i))$ differ from $b_{2, i} = B(x_i, M_2(x_i))$ on the test data used in the experiments? What about standard distance functions, summary statistics, tests or classifiers defined over the test $b$'s? I'd recommend including these types of empirical studies given that: a) the primary contribution of the paper seems to be a novel application of DAVT for change detection in models wrt `B`'s outputs and b) it's increasingly common practice to test a release candidate using the outputs of multiple ML model-based "test" or "correctness" oracles (i.e., different `B`'s, trained separately) on held-out test and regression data.

- Overall, given the current experimental section's weaknesses, I'm somewhat concerned that the method might be at risk of not outperforming existing methods of large model evaluation and being of limited utility to practitioners. As a result, I'm leaning towards rejection. That said, I don't think there's anything technically wrong with the paper and it's possible I may have missed something obvious. I'm open to revising my score upwards and look forward to the author response and other reviews.

**Questions:**

1. How does the proposed method handle change in `B` itself (from updates, prediction error, concept drift)?

2. How does the proposed method handle multiple behaviors (low toxicity and low hallucinations and ...)?

3. Is it possible to construct a baseline by simply comparing `B`'s values on the two models' outputs using test data? See above for details. If yes, how does it compare with the proposed method?

---

> ### Author Response · Authors · 2024-11-19
> **We discuss our method's sensitivity to the behavior scoring function, an extension to multiple behaviors and baseline comparisons.**
>
> * *[..it is reasonable to expect large-scale implementations of $B$ to involve an ML model and therefore add noise…A deeper discussion and empirical investigation into the sensitivity of the proposed method wrt different types of $B$'s would strengthen the paper..]*
>   * Thank you for this. We agree that the design of accurate behavior scoring functions can be difficult and will likely involve ML models. To highlight this we have added a paragraph discussing the difficulties involved in using a behavior scoring function in Section 6.
>   * Our proposed test for Behavioral Shift Auditing (BSA) is robust to some noise in the behavior scoring function. Specifically, as long as the expected output of the noisy scoring function corresponds to the true behavior score – that is, $B(\mathbf{x,y})=\mathbb{E}[\tilde{B}(\mathbf{x,y})]$ for every $(\mathbf{x,y})$, where $\tilde{B}$ is the stochastic behavior scoring function and $(\mathbf{x,y})$ denotes a prompt-generation pair – the statistical guarantees of our test remain valid in the limit.
>   * However, we agree that an empirical investigation into the effects of a noisy scoring function is important as the noise might make it harder for the betting score network to learn. To investigate this, we conducted an experiment (summarized in Appendix C.4) where we added varying levels of Gaussian noise to behavior scores. The results show that it takes our test longer to detect a distributional difference the noisier the behavior scoring function is, but also that detection rates still converge to the same level. This sensitivity study gives us insight into the robustness of the proposed test for BSA, thank you for suggesting this.
>   * In Appendix C.4, we have also added a deeper discussion about the effects of certain types of systematic errors in the behavior scoring function on test results.
>
> * *[..a practitioner might consider it insufficient for the model to pass a **single** test of behavior…It's unclear how the proposed method would handle multiple behavior objectives (e.g., low toxicity and low hallucinations and ...)..]*
>   * We agree that a practitioner might want to test for changes in multiple behaviors, and a useful prompt dataset would exhibit each behavior to be tested (so that both the propensity to hallucinate and toxicity could be evaluated in the model generations). Our test can be naturally extended to this case:
>    * For the **exact test**, the extension is straightforward, as it is an application of Deep Anytime-Valid Testing (DAVT), which has been successfully applied to multi-dimensional distributions [1].
>    * For the **tolerance test**, one can either set a *global* tolerance hyperparameter as the maximally-allowed distance between multidimensional distributions, or conduct multiple tests in parallel with individual thresholds for each behavior. We discuss this extension in Appendix C.5 and briefly mention it in Section 6 of the main text.
>
> * *[..Is it possible to construct a baseline by simply comparing $B$'s values on the two models' outputs using test data? ..]*
>   * Thank you for this. We had a similar difficulty finding a baseline for this exact problem. Our thinking was that since our work is an extension of Deep Anytime-Valid Testing (DAVT), and they have conducted a detailed comparison with other testing baselines and found their test to generally outperform those baselines, that this was the most informative comparison [1].
>   * Because of this we focussed our effort on sensitivity studies for BSA.
>   * We also report mean toxicity scores for different LMs in Figure 8 in Appendix A.2 and mean BLEU scores in the main text, Section 4.2 line 416/417 (“Few-shot prompting leads to a modest increase in mean BLEU scores from 0.1683 to 0.1765.”). Wasserstein distances between instruction-tuned checkpoints and baselines can be found in Figure 2 and 4, as well as Figure 10 in Appendix C.1.
>   * However, while **summary statistics** (e.g., mean and variance) and **distance measures** (e.g., Wasserstein distance) are commonly used to compare distributions, they may not capture critical aspects of behavior distributions or provide clear decision rules for detecting changes.
>   * For completeness we have also added new results evaluating the classical two-sample Kolmogorov-Smirnov (KS) test. Unlike our anytime-valid method, classical tests such as this can suffer from inflated Type I error rates when used repeatedly as new data is added. This can lead to increased false positives and unreliable conclusions. In Appendix C.3, we compare our test to the KS test and show experimentally that our method avoids the pitfalls of increased Type I errors, highlighting its robustness and practical advantages over traditional methods.
>
> [1] Pandeva, T., Forré, P., Ramdas, A., & Shekhar, S. (2024, April). Deep anytime-valid hypothesis testing. In International Conference on Artificial Intelligence and Statistics (pp. 622-630). PMLR.

---

> ### Author Response · Authors · 2024-11-21
>
> Dear Reviewer Y3yQ,
>
> Thank you for your thoughtful feedback on our submission. We have tried to carefully address each point raised in our response and we have made corresponding improvements to the paper. We would be grateful if you could review our revisions and responses to your comments. If you have any remaining questions or concerns, we welcome your additional feedback.
>
> -The Authors

---

> > ### Author Response · Authors · 2024-12-01
> >
> > Dear Reviewer Y3yQ,
> >
> > We thank the reviewer for the helpful comments and the opportunity for further discussion. As we are approaching the end of the review process, could you kindly let us know if there are any other concerns that we should address?
> >
> > -The Authors

---

### Meta-Review · Area_Chair_fGk4 · 2024-12-20

**Metareview:**

The authors consider audits on identifying changes to the behavior of an LM over time, using ideas from statistical hypothesis testing to identify precisely when behaviors shift. They assume the existence of some measurable 'behavior' (e.g. toxicity classifier) that they want to track, and flag when the distribution of responses on that behavior differs.

Transparency into the behaviors of LLMs is important, and doing so in a precise, statistically rigorous way is useful when the models themselves can be stochastic and hard to test. There is also a nice anytime valid test that provides control despite the optional stopping problems that arise in these types of problems.

As a weakness, reviewers brought out issues with the validity of the behavior model, as well as some writing comments.

**Additional Comments On Reviewer Discussion:**

The authors responded very extensively to various limitations questions addressed by the reviewers. Changes to the score were minor, but the discussion process was helpful in understanding the full scope of limitations.

---

### Decision · Program_Chairs · 2025-01-22

Accept (Poster)